# Independent tuning of size and coverage of supported Pt nanoparticles using atomic layer deposition

Jolien Dendooven [1], Ranjith K. Ramachandran [1], Eduardo Solano[1,7], Mert Kurttepeli[2], Lisa Geerts [3], Gino Heremans [3], Jan Rongé [3], Matthias M. Minjauw [1], Thomas Dobbelaere[1], Kilian Devloo-Casier[1], Johan A. Martens[3], André Vantomme[4], Sara Bals[2], Giuseppe Portale [5,8], Alessandro Coati[6] & Christophe Detavernier [1]

Synthetic methods that allow for the controlled design of well-defined Pt nanoparticles are highly desirable for fundamental catalysis research. In this work, we propose a strategy that allows precise and independent control of the Pt particle size and coverage. Our approach exploits the versatility of the atomic layer deposition (ALD) technique by combining two ALD processes for Pt using different reactants. The particle areal density is controlled by tailoring the number of ALD cycles using trimethyl(methylcyclopentadienyl)platinum and oxygen, while subsequent growth using the same Pt precursor in combination with nitrogen plasma allows for tuning of the particle size at the atomic level. The excellent control over the particle morphology is clearly demonstrated by means of in situ and ex situ X-ray fluorescence and grazing incidence small angle X-ray scattering experiments, providing information about the Pt loading, average particle dimensions, and mean center-to-center particle distance.

[1] Department of Solid State Sciences, COCOON, Ghent University, Krijgslaan 281/S1, B-9000 Ghent, Belgium. [2] Electron Microscopy for Materials Science (EMAT), University of Antwerp, Groenenborgerlaan 171, B-2020 Antwerp, Belgium. [3] Center for Surface Chemistry and Catalysis, KU Leuven, Celestijnenlaan 200F, B-3001 Leuven, Belgium. [4] Instituut voor Kern- en Stralingsfysica, KU Leuven, Celestijnenlaan 200D, B-3001 Leuven, Belgium. [5] ESRF European Synchrotron, DUBBLE Beamline BM26, Avenue des Martyrs, CS40220, 38043 Grenoble, France. [6] Synchrotron SOLEIL, SixS Beamline, L'Orme des Merisiers, Saint-Aubin, BP48, 91192 Gif-sur-Yvette, France. [7] Present address: ALBA Synchrotron Light Source, NCD beamline, Carrer de la Llum 2-26, 08290 Cerdanyola del Vallès, Spain. [8] Present address: Macromolecular Chemistry & New Polymeric Materials, University of Groningen, Nijenborgh 4, AG Groningen, 9747, The Netherlands. Correspondence and requests for materials should be addressed to C.D. (email: Christophe.Detavernier@UGent.be)

Supported Pt nanoparticles and nanoclusters occupy an important position in heterogeneous catalysis and electro-catalysis, e.g., for CO/NO$_x$ oxidation in catalytic converters, hydrocarbon reforming, production of nitric acid, the oxygen reduction reaction (ORR) in fuel cells, and the hydrogen evolution reaction (HER) in water electrolysis[1–5]. It is well established that the activity and selectivity of catalytic nanoparticles is closely related to their size and shape, while also the nanoparticle coverage, and in turn, the center-to-center particle distance, will affect the performance of supported catalysts[6, 7]. Understanding the relation between catalytic performance and structural properties of nanocatalysts is a major goal in fundamental catalysis research and is of critical importance toward a more rational design of catalysts with improved activity and selectivity. In this regard, there is a high interest in synthesis methods that offer control at the atomic level and enable the fabrication of well-defined supported nanoparticles with narrow size distributions. Indeed, conventional methods such as wet impregnation or precipitation often yield a wide distribution of particle sizes and lack independent control of size and coverage. In recent years, significant progress has been achieved in the solution-phase synthesis of monodisperse ligand-protected nanoparticles[8, 9], but it remains challenging to remove the ligands after deposition on the support without causing particle aggregation and size enlargement[10]. Another approach is based on the soft deposition of well-defined metal clusters, prepared and mass-separated in the gas phase[11]. However, even though subnanometer dimensions can be achieved, the method is usually limited to low particle coverages in order to avoid cluster aggregation, and deposition onto non-planar high surface area supports remains challenging[12].

In the past few years, atomic layer deposition (ALD) has gained attention as a versatile technique for the synthesis of nanoscale catalysts[13–18]. ALD relies on sequential self-limiting reactions between gas-phase precursor molecules and a solid support to deposit a variety of materials including several oxides, nitrides, sulfides, and (noble) metals[19]. The surface-controlled chemistry of ALD yields a sub-nanometer thickness control and an excellent conformality on large surface area supports; two benefits that are extremely relevant toward the atomic-scale design of catalysts. In addition, many noble metal ALD processes are characterized by a nucleation-controlled growth, wherein nanoparticles are formed at the start of the deposition instead of continuous layers. This has led to several ALD-based strategies for the synthesis of monometallic and bimetallic nanoparticles[14, 16, 18]. Well-chosen combinations of noble metal ALD chemistries and processing conditions can result in alloyed[20–22] or core/shell[23, 24] nanoparticles containing Pt, Pd, and/or Ru. Another approach consisted of using metal oxide ALD prior to Pd deposition to modify the number of nucleation sites and hence nanoparticle loading[25]. Metal oxide ALD overcoats have also proven very effective to stabilize metal nanoparticles in high temperature reactions while preserving their catalytic activity[26, 27].

Catalytically active Pt nanoparticles have successfully been synthesized by ALD on large surface area carbon[28] and oxide[29–32] supports and metal-organic frameworks[33]. Generally, narrow size distributions were obtained, and the Pt loading and average particle size were shown to increase with the number of applied ALD cycles. However, the evolution in particle coverage is often not investigated, nor is independent tuning of size and coverage demonstrated.

Most of the aforementioned studies employed the standard ALD process for Pt that was originally developed by Aaltonen et al. and uses trimethyl(methylcyclopentadienyl)platinum (MeCpPtMe$_3$) as precursor and O$_2$ as reactant at a temperature of ca. 300 °C[34]. Over the past decade, O$_2$ plasma[35–37], O$_3$[38], H$_2$[39], H$_2$ plasma[40], N$_2$ plasma[37], and NH$_3$ plasma[37] were also

demonstrated to be effective reactants for Pt ALD from the MeCpPtMe$_3$ precursor. The choice of reactant is expected to influence the nucleation and island growth, thus offering a potential control knob for tuning the synthesis of Pt nanoparticles.

Grazing incidence small angle X-ray scattering (GISAXS) is a synchrotron-based morphological characterization technique[41] that requires no special sample preparation and is therefore ideally suited for a variety of in situ experiments, including the characterization of deposition processes in high vacuum[42]. The technique has, for example, been used for real-time monitoring of noble metal growth by evaporation[43] and sputtering[44–47]. We recently designed a mobile setup for synchrotron-based in situ characterization during both thermal and plasma-enhanced ALD[48].

In this work, in situ GISAXS and X-ray fluorescence (XRF) characterizations are applied during Pt ALD to elucidate how the average particle dimensions and mean center-to-center particle distance evolve with increasing Pt loading. The results reveal a clearly different particle growth behavior for the N$_2$ plasma-based (N$_2^*$-based)[37] Pt ALD process compared to the standard O$_2$-based[34] process. Moreover, by combining these two ALD processes, we provide a strategy for independent tuning of the Pt nanoparticle size and areal density, expanding the potential of the ALD technique for the synthesis of well-defined Pt nanoparticles with exciting opportunities for fundamental catalysis research.

## Results

**O$_2$-based vs. N$_2$ plasma-based ALD of Pt nanoparticles.** The Pt nanoparticle growth is studied on planar Si substrates covered with native SiO$_2$. The Pt depositions are performed in an experimental ALD setup equipped with Be windows and positioned in the path of a 12.75 keV X-ray beam at the SixS beamline of the SOLEIL synchrotron radiation facility (Saint-Aubin, France)[48]. Prior to ALD growth, the Si/SiO$_2$ support is heated to 300 °C and exposed to oxygen plasma to clean the sample surface and improve the reproducibility between different experiments. Pt is deposited at 300 °C using an ALD cycle comprising 15 s exposure to the MeCpPtMe$_3$ precursor at ca. 1 mbar followed by either 10 s exposure to O$_2$ at ca. 1 mbar or two times 10 s exposure to N$_2^*$ at ca. $10^{-2}$ mbar. In between the exposures, the chamber is evacuated to a pressure of $10^{-6}$ mbar and every two ALD cycles XRF and GISAXS data are acquired (measurement details may be found in the Methods section). The integrated counts for the Pt L$\alpha$ XRF peak are converted to a surface density of Pt atoms based on a calibration of the XRF spectrometer against Rutherford backscattering measurements. The evolution of the Pt loading with the number of deposition cycles is depicted in Supplementary Fig. 1 and is marked by a slow growth rate during the initial ALD cycles, as expected for a nucleation-controlled growth process. For Pt loadings of 45 atoms per nm$^2$ and higher (>16 O$_2$ or 14 N$_2^*$-based Pt ALD cycles), the presence of Pt nanoparticles on the SiO$_2$ surface is accompanied by the appearance of a clear scattering peak in GISAXS at $q_y \neq 0$ nm$^{-1}$ (Fig. 1a). This peak arises because the Pt islands are not randomly distributed, but their arrangement on the surface is characterized by a mean center-to-center distance $D$. For the O$_2$-based Pt ALD process, the main scattering maximum gradually shifts to lower $q_y$-values with progressing deposition of Pt atoms (Fig. 1a, left), suggesting an increase in particle distance $D$. In contrast, for the N$_2^*$-based Pt ALD process, the position of the main scattering peak and thus also the distance $D$ remain constant during the deposition (Fig. 1a, right).

Analysis of the GISAXS data through comparison with simulations assumes a certain shape for the Pt nanoparticles.

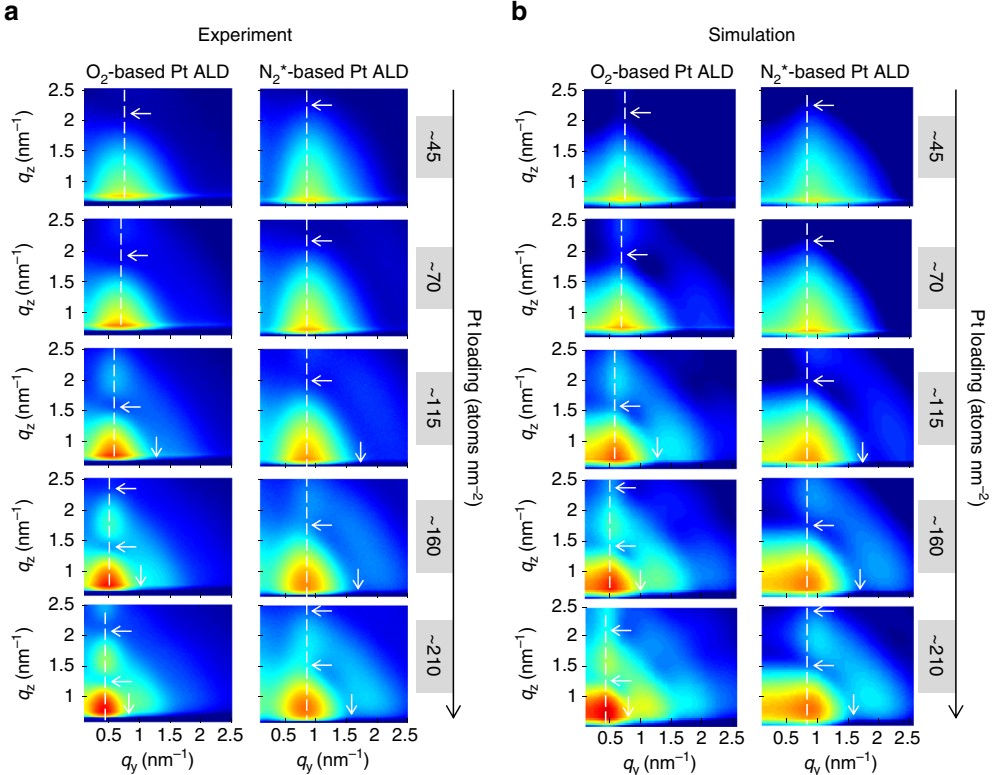

**Fig. 1** In situ morphological characterization during Pt ALD processes. **a** Selection of experimental 2D GISAXS images measured in situ during (left) $O_2$-based Pt ALD and (right) $N_2$*-based Pt ALD. **b** Corresponding simulated patterns. All details about the calculations may be found in Supplementary Note 2 and Supplementary Figs. 7 and 8. The dashed vertical lines indicate the $q_y$ position of the main scattering lobe. The horizontal/vertical arrows indicate the minima along the $q_z/q_y$ direction

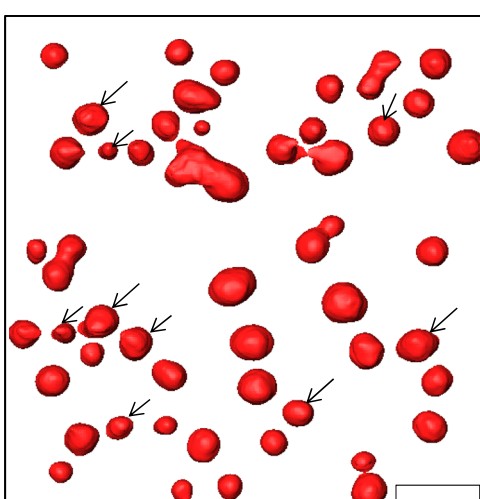

**Fig. 2** Electron tomography characterization of the Pt nanoparticle shape. 3D volume rendering, viewed from the top, of Pt nanoparticles synthesized with the $O_2$-based ALD process (Pt loading ~45 atoms per $nm^2$). The arrows indicate particles that exhibit a flat surface at the Pt/$SiO_2$ interface. The scale bar indicates 10 nm

Therefore, to obtain more insights in the three-dimensional (3D) shape of the nanoparticles, an electron tomography study is performed on a sample with a Pt loading of ~45 atoms per $nm^2$ prepared by the $O_2$-based ALD process. While conventional transmission electron microscopy (TEM) only yields a two-dimensional (2D) projection of a 3D object, electron tomography allows reconstructing the 3D structure of the object based on a

large number of 2D projection images[49]. To acquire a full tilt range of 2D projection images with high angle annular dark field (HAADF) scanning transmission electron microscopy (STEM), a plan-view sample is prepared as explained in the Methods section and mounted on a dedicated tomography holder. After acquisition and alignment of the HAADF-STEM images, the "Simultaneous Iterative Reconstruction Technique" is used for the reconstruction of the 3D structure of the specimen. The reconstructed volume of Pt nanoparticles deposited in a ca. 60 by 60 $nm^2$ region on the Si/$SiO_2$ surface is visualized in Fig. 2. An animated version of the tomogram is provided in Supplementary Movie 1. The majority of the Pt clusters have a spheroidal shape, while some clusters consist of agglomerated smaller particles. Rounded particles are expected to expose many atomic steps and kinks presenting high catalytic activity[50, 51]. In the animated version of the tomogram, it can also be observed that some of the particles exhibit a flat surface at the Pt/$SiO_2$ interface (particles indicated by an arrow in Fig. 2), while others seem to be full spheroids. The tomography study thus suggests that the contact angle of the Pt nanoparticles with the surface varies from particle to particle. However, it should be kept in mind that the tomography series was carried out on a plan-view TEM sample as indicated previously. Due to missing wedge artifacts[49], which are more pronounced at the interface between the nanoparticles and the substrate in the case of plan-view direction, the real morphology of the nanoparticles might vary slightly.

Inspired by the work of Schwartzkopf et al.[44], a GISAXS analysis strategy is developed to obtain cycle-per-cycle information about the average particle distance and dimensions. The analysis approach is extensively explained in Supplementary Note 1. The obtained morphological parameters for the $O_2$-based and $N_2$*-based Pt ALD processes are compared in Fig. 3a. To

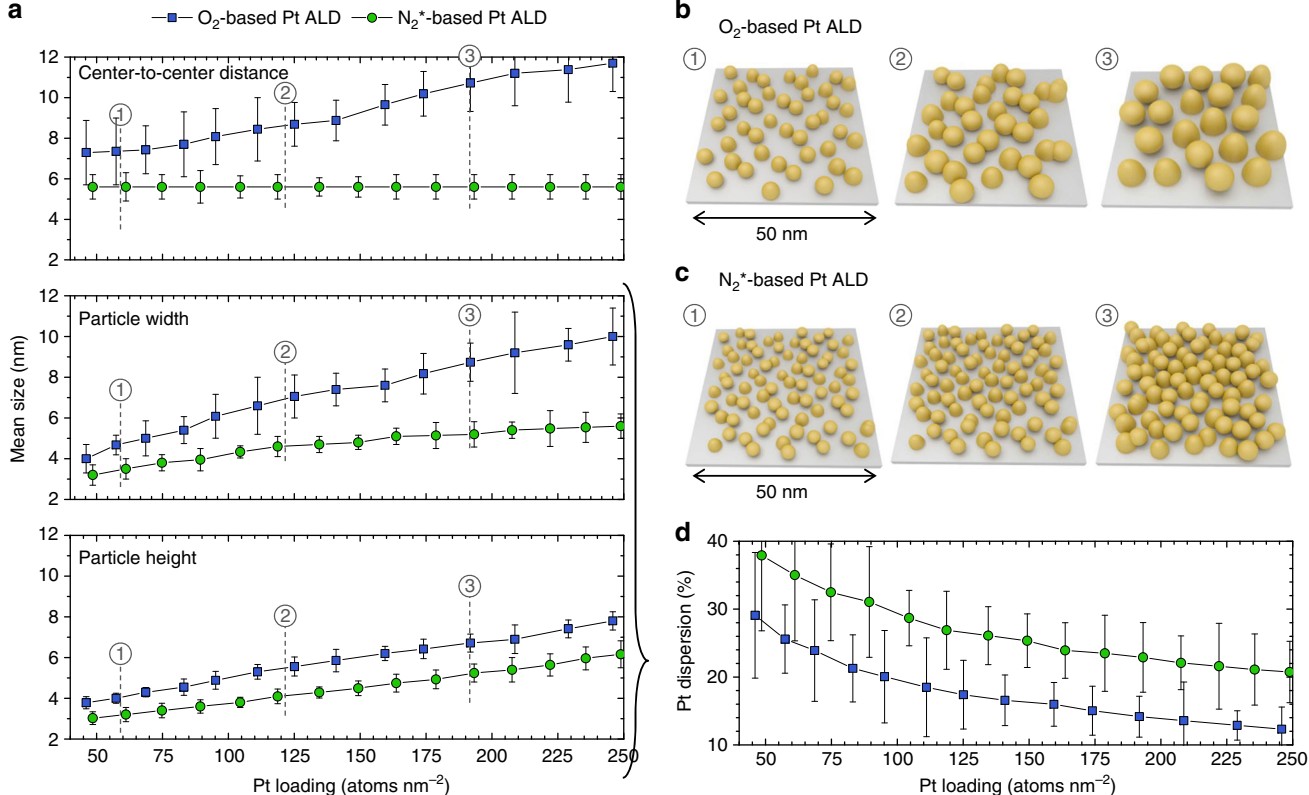

**Fig. 3** Morphological evolution of Pt nanoparticles during ALD. **a** In situ data, obtained from GISAXS and XRF, on mean center-to-center distance (top), particle width (middle) and particle height (bottom) against Pt loading for $O_2$-based Pt ALD (blue squares) and $N_2^*$-based Pt ALD (green circles). The error bars represent the uncertainties of the obtained values. These uncertainties were estimated based on a quantitative comparison of experimental and calculated 1D vertical (for the particle height) and horizontal (for the particle width and center-to-center distance) GISAXS line profiles. The sum of squared residuals (SSR) was calculated for varying input values for one of the parameters (while the other parameters were kept constant). The presented uncertainties correspond to an increase in the SSR of ca. 50%. All details about the calculations may be found in Supplementary Note 3. **b**, **c** Schematic representation of the GISAXS results for Pt loadings of (1) ~60, (2) ~120, and (3) ~190 Pt atoms per $nm^2$ obtained using $O_2$-based Pt ALD (**b**) and $N_2^*$-based Pt ALD (**c**). **d** Pt dispersion, i.e., fraction of accessible Pt atoms, calculated from the particle dimensions and shape, as obtained from GISAXS, against Pt loading for $O_2$-based Pt ALD (blue squares) and $N_2^*$-based Pt ALD (green circles). The error bars were derived from the uncertainties of the particle dimensions and error propagation theory

validate the analysis strategy, the experimental GISAXS images are compared with simulated patterns obtained with the software IsGISAXS[52]. These simulations assume a spheroidal shape for the Pt nanoparticles and use, amongst other parameters, the average center-to-center distance, particle width and particle height obtained from the analysis as input (all details can be found in Supplementary Note 1). Figure 1b shows the simulations that correspond to the experimental data in Fig. 1a. As indicated by the white arrows and dashed lines, the positions of the different maxima/minima are successfully reproduced, justifying our analysis strategy to derive the average morphological parameters and confirming the spheroidal shape of the nanoparticles. As exemplified in Supplementary Note 1 and Supplementary Fig. 6, best agreement with the experimental GISAXS patterns is obtained when a two particle model is used to describe the spheroidal particles[53]. A mixture of 50% (75%) full spheroids and 50% (25%) hemi-spheroids is assumed to simulate the patterns for the $O_2$-($N_2^*$-)based ALD process. This mixture of two different wetting conditions in the simulations suggests a real situation where the contact angle of the spheroidal Pt nanoparticles with the $SiO_2$ surface varies from particle to particle, as also suggested by the tomography result. Moreover, GISAXS indicates (on average) larger contact angles (larger dewetting) for the Pt nanoparticles deposited via the $N_2^*$-based ALD process.

As indicated before, the obtained morphological parameters for the $O_2$-based and $N_2^*$-based Pt ALD processes are compared in Fig. 3a. Real-space sketches of the morphology at specific Pt loadings are shown in Fig. 3b and c, respectively. The different evolution in the mean center-to-center particle distance $\langle D \rangle$, and thus particle coverage, estimated as $1/\langle D \rangle^2$, for the two processes is confirmed in Fig. 3a, top (blue squares vs. green circles) and is clearly observed in the sketches. The results further reveal a higher lateral growth rate for the $O_2$-based process than for the $N_2^*$-based process (blue squares vs. green circles in Fig. 3a, middle). Moreover, comparison of the average particle width and height for the $O_2$-based process (blue squares in Fig. 3a, middle vs. bottom) reveals the formation of laterally elongated nanoparticles. The continuously increasing particle distance and enhanced lateral growth both point to a diffusion-mediated growth regime during $O_2$-based Pt ALD[44]. Ripening of the Pt nanoparticles is likely induced by the adsorption of mobile $PtO_x$ species that are formed during the $O_2$ exposure in the second ALD half cycle[17, 54, 55]. In contrast, in $N_2^*$-based Pt ALD, atom and cluster surface diffusion seem to be suppressed, leading to a static particle growth. For Pt loadings in the range 45–250 atoms per $nm^2$, the $N_2^*$-based process results in particle geometries with a ca. 1:1 aspect ratio (green circles in Fig. 3a, middle vs. bottom).

Knowing the geometrical dimensions of the Pt nanoparticles from GISAXS, we also estimate the dispersion, or fraction of

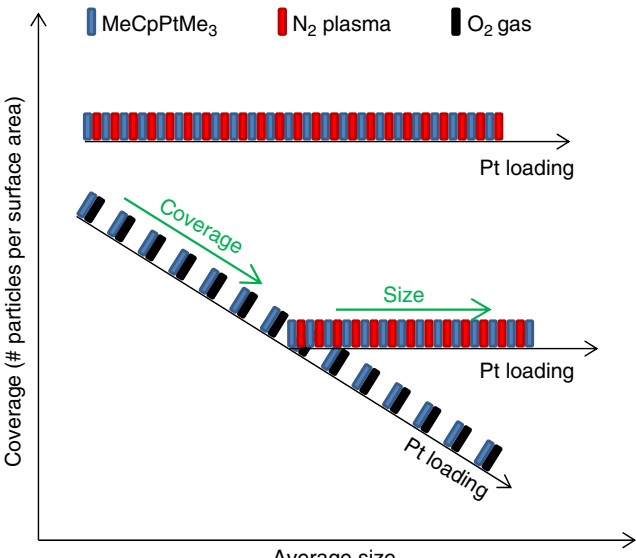

**Fig. 4** Morphological tuning of Pt nanoparticles using ALD. ALD recipe scheme showing the evolution of Pt nanoparticle coverage and size during the application of $O_2$-based and $N_2^*$-based Pt ALD cycles. The blue, red, and black rectangles represent exposures to the Pt precursor, $N_2$ plasma and $O_2$ gas, respectively. For the $O_2$-based Pt ALD process (alternating blue and black rectangles), the coverage decreases and the average size increases with increasing number of ALD cycles. For the $N_2^*$-based Pt ALD process (alternating blue and red rectangles), the coverage remains constant and the average size increases with increasing number of ALD cycles. By combining both processes, both the particle coverage and particle size can be tuned as indicated by the green arrows

accessible Pt atoms, as a function of the Pt loading (Fig. 3d). The dispersion is calculated by dividing the number of atoms available at the surface of a Pt particle (multiplied by a factor of 5/6 to account for the inaccessibility of the atoms located at the particle/support interface)[56, 57] by the total number of Pt atoms in one particle. As expected, the dispersion decreases with increasing Pt loading for both ALD processes. However, a 10% larger dispersion is obtained with the $N_2^*$-based Pt ALD process compared to the standard $O_2$-gas process.

**ALD strategy for tuning Pt nanoparticle size and coverage**. In previous studies, tuning of the Pt nanoparticle size has often been achieved by increasing the number of ALD cycles using the standard $O_2$-gas ALD process[17, 29–32]. However, the results presented in Fig. 3a show that for this process not only the size increases with the number of ALD cycles (or Pt loading) but also the center-to-center distance $\langle D \rangle$, implying a decrease in particle areal density. As schematically represented in the ALD recipe scheme in Fig. 4, both size and coverage are thus modified simultaneously when using the $O_2$-based ALD process. In contrast, independent size tuning at fixed particle coverage can be achieved using $N_2^*$-based Pt ALD. However, control over both particle coverage and size cannot be achieved by varying the number of ALD cycles of one process type and, therefore, we propose a tuning strategy that combines the $O_2$-based and $N_2^*$-based ALD processes to enable the synthesis of supported Pt nanoparticles with independent control over the coverage and size. By first applying $O_2$-based Pt ALD, the center-to-center distance between the particles can be controlled. Subsequently, the size of the particles can further be tailored by applying $N_2^*$-based Pt ALD.

This approach is verified by performing 24 cycles of the $O_2$-based ALD process followed by $N_2^*$-based ALD cycles and monitoring the growth in situ with GISAXS and XRF. Figure 5a shows a selection of 2D GISAXS patterns measured during this process. The evolution in $q_y$-position of the main scattering peak is clearly visible in the 2D color plot representing the horizontal line profiles at maximum intensity (Fig. 5b). As expected, the maximum of the scattering peak shifts to lower $q_y$-values during the $O_2$-based ALD growth while the peak position remains constant during subsequent growth with the $N_2^*$-based Pt ALD process. The GISAXS and XRF analysis results for the combined process are depicted in Fig. 5c (yellow triangles) together with the earlier discussed results obtained for the pure $O_2$-based and $N_2^*$-based Pt ALD processes (blue squares and green circles, respectively; GISAXS analysis details and simulated patterns may be found in Supplementary Note 2 and Supplementary Fig. 9). This graph displays the average particle coverage $1/\langle D \rangle^2$ (left $y$-axis) and mean center-to-center distance $\langle D \rangle$ (right $y$-axis) against the average particle size in number of Pt atoms (bottom $x$-axis). The latter is calculated by dividing the Pt loading by the particle areal density $1/\langle D \rangle^2$. The top $x$-axis shows the diameter of a spherical particle with a 1:1 aspect ratio and equivalent volume. During the $O_2$-based Pt ALD cycles, the coverage decreases, reaching a value of $\sim 1.6 \times 10^{12}\,\text{cm}^{-2}$ at cycle 24, and the size increases, reaching an equivalent diameter of $\sim 5.2$ nm at cycle 24. Figure 5d(1) shows a representative SEM image of this morphology together with a schematic representation of the GISAXS result. As discussed above, continued growth with the $O_2$-based process results in an increase in particle size accompanied by a decrease in particle coverage due to dynamic particle coalescence (blue squares in Fig. 5c). For high Pt loadings >250 atoms per nm$^2$, worm-like features are obtained for this process instead of isolated particles as observed in SEM (Fig. 5d (2)). In contrast, when 24 $O_2$-based ALD cycles are followed by $N_2^*$-based Pt ALD, the particles grow in size while the coverage remains constant at $\sim 1.6 \times 10^{12}\,\text{cm}^{-2}$ (fixed peak position in Fig. 5b and yellow triangles in Fig. 5c). SEM confirms the formation of isolated particles, even for Pt loadings >350 atoms per nm$^2$ (Fig. 5d(3)).

The in situ data set confirms that independent control over coverage and size can be achieved by combining $O_2$-based and $N_2^*$-based Pt ALD. The region marked in orange in Fig. 5c shows the coverage-size combinations that can be achieved by this tuning approach. When targeting a specific particle size, the minimum/maximum particle coverage is obtained when solely $O_2/N_2^*$-based Pt ALD is applied (blue squares/green circles in Fig. 5c). To reach a particle coverage in between these extremes, one applies the $O_2$-based ALD process until the targeted coverage is reached, and then continues with the $N_2^*$-based ALD process until the targeted particle size is obtained (resulting in a growth trajectory similar to the yellow triangles in Fig. 5c).

**Tuning coverage during ALD of Pt nanoparticles**. To demonstrate the level of precision and the range over which the particle coverage can be tuned, a series of depositions are performed where the total number of ALD cycles is kept constant at 60, while the number of $O_2/N_2^*$-based Pt ALD cycles is increased/decreased from 0/60 (sample A) over 20/40 (sample B) and 30/30 (sample C) to 40/20 (sample D). The considered ALD recipes are schematically depicted in the coverage/size diagram in Fig. 6a. The first deposition is equivalent to the green circles in Fig. 5c, while the three other depositions are expected to result in growth trajectories similar to the yellow triangles. XRF analysis revealed a similar Pt loading in the range of 190–210 Pt atoms per nm$^2$ for all samples. The GISAXS patterns, recorded ex situ at the

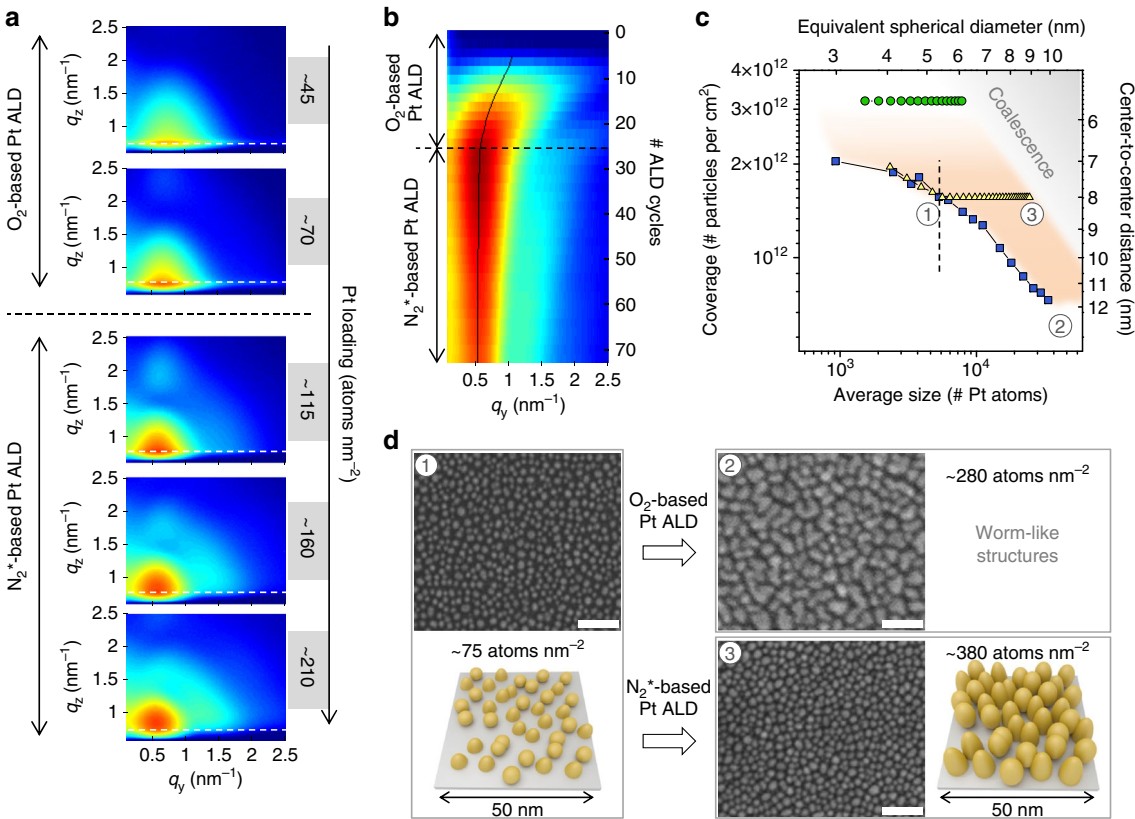

**Fig. 5** In situ characterization of the morphological tuning of Pt nanoparticles during ALD. **a** Selection of 2D GISAXS images measured in situ during a combined process using first 24 cycles of the $O_2$-based Pt ALD process followed by $N_2^*$-based Pt ALD. **b** 2D color map showing the intensity evolution of the line profiles at the Si Yoneda position indicated by the dashed horizontal lines in **a**. **c** Average particle coverage against particle size for $O_2$-based Pt ALD (blue squares), $N_2^*$-based Pt ALD (green circles), and the combined process using first 24 cycles of the $O_2$-based Pt ALD process (left side of the dashed line) to tune the particle coverage followed by $N_2^*$-based Pt ALD (right side of the dashed line) to tune the particle size (yellow triangles). **d** Representative SEM images and schematic representations of the GISAXS results after (1) 24 $O_2$-based ALD cycles, (2) continued growth with the $O_2$-based ALD process, and (3) continued growth with the $N_2^*$-based ALD process. The white scale bars indicate 50 nm

DUBBLE BM26B beamline[58] of the European synchrotron radiation facility (ESRF, Grenoble, France) and depicted in Fig. 6b, the schematic representation of the GISAXS analysis results in Fig. 6c and the SEM images in Fig. 6d corroborate the capability of tuning the particle areal density (GISAXS analysis details and simulated patterns may be found in Supplementary Note 2 and Supplementary Fig. 10). As before, the shift in GISAXS lobe position to lower $q_y$-values indicates the increase in mean center-to-center particle distance, and thus the decrease in particle coverage, with increasing number of $O_2$-based Pt ALD cycles. The mean center-to-center distances $\langle D \rangle$ and corresponding estimated particle coverages $1/\langle D \rangle^2$ evolve from ~5.6 nm/3.2×10^{12} cm^{-2} (sample A) over ~7.1 nm/2.0×10^{12} cm^{-2} (sample B) and ~8.7 nm/1.3×10^{12} cm^{-2} (sample C) to ~10.8 nm/8.6×10^{11} cm^{-2} (sample D). Comparison of the particle width and height indicates an enhanced lateral growth with increasing number of $O_2$-based Pt ALD cycles (sketches in Fig. 6b, middle), as expected based on the observations made for the pure $O_2$-based process (blue squares in Fig. 3a). The average particle widths obtained from GISAXS are in good agreement with particle size distribution analyses of the SEM images (Supplementary Note 4; Supplementary Fig. 13).

**Tuning size during ALD of Pt nanoparticles**. The size tuning capability is demonstrated by a second series of experiments where the number of $O_2$-based Pt ALD cycles is held constant at

20 and followed by 0 (sample I), 20 (sample II), 30 (sample III), or 40 (sample IV) $N_2^*$-based Pt ALD cycles (see ALD recipe diagram in Fig. 7a). The amount of deposited Pt atoms increases as expected with increasing number of ALD cycles (Fig. 7c). The fixed position of the main scattering lobe in the ex situ GISAXS patterns (Fig. 7b) and inspection of the SEM images (Fig. 7d) confirm that the particle areal density is determined by the 20 $O_2$-based Pt ALD cycles and is equal for all samples. Indeed, quantitative analysis of the SEM images for samples II, III, and IV by manually counting the Pt nanoparticles in a 150 by 150 nm^2 area yields particle coverages of 1.58×10^{12} cm^{-2}, 1.60×10^{12} cm^{-2}, and 1.53×10^{12} cm^{-2}, respectively. These values correspond to an average center-to-center distance $\langle D \rangle$ of 8.0 ± 0.1 nm. This value is higher than the one obtained from GISAXS analysis, 7.1 nm, which is likely due to the fact that agglomerated particles and particles at the edges of the SEM images are excluded from the particle count. For sample I, the contrast between the background and the small nanoparticles in SEM is insufficient to allow for a reliable particle count.

On the other hand, the GISAXS patterns show clear differences with respect to the positions of the secondary intensity lobes. The horizontal arrows in Fig. 7b point to the minima along the $q_z$-direction and the decrease in $q_z$-value of these minima is indicative of an increase in particle height. Similarly, the shift of the intensity minimum along the $q_y$-direction to lower $q_y$-values (vertical arrow in Fig. 7b, left) is related to an increase in the lateral size of the Pt particles. The results of the quantitative

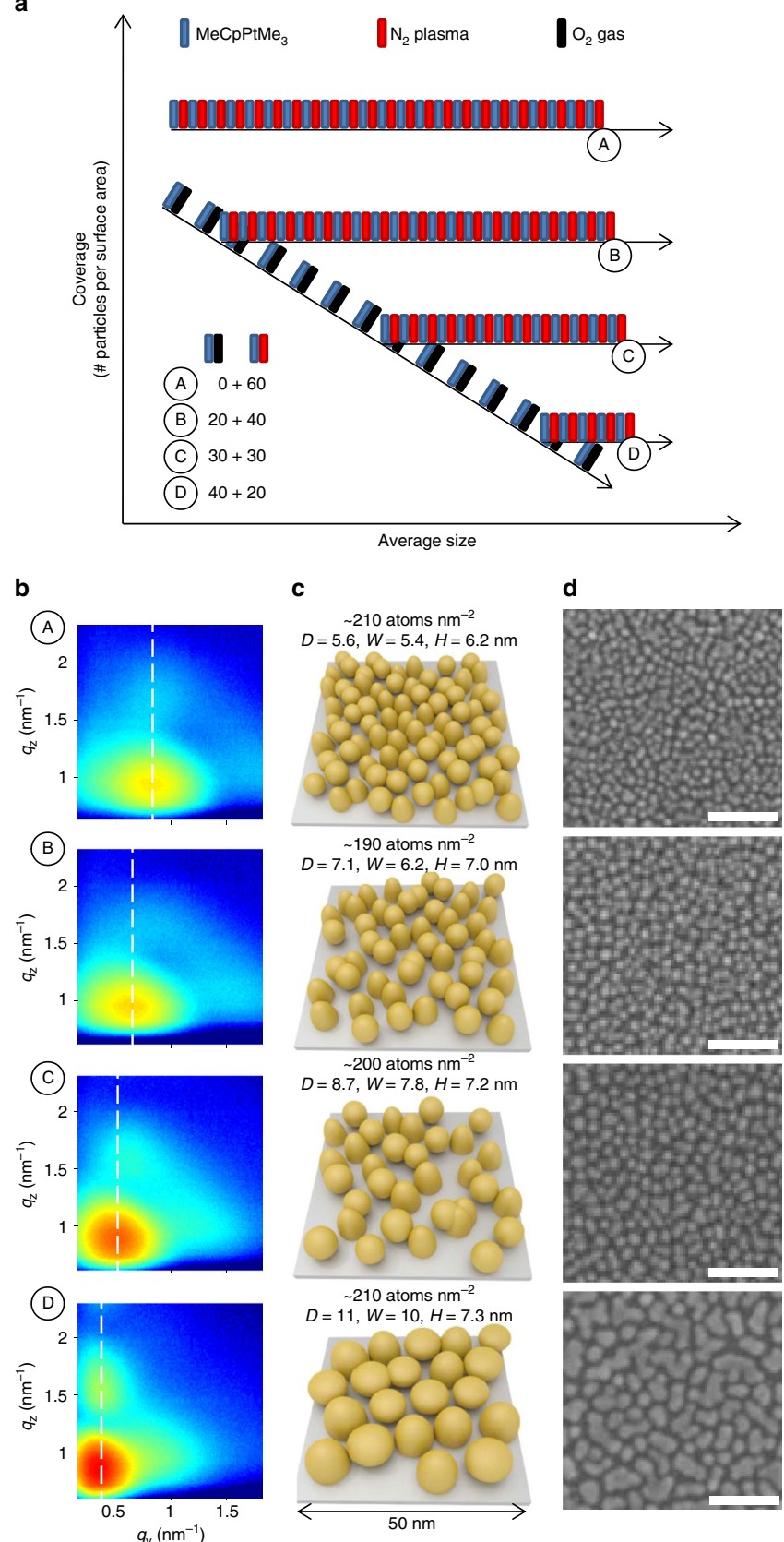

**Fig. 6** Precise control of coverage of Pt nanoparticles using ALD. Tuning the particle coverage at a fixed Pt loading by combining 0 (sample A), 20 (sample B), 30 (sample C), and 40 (sample D) $O_2$-based Pt ALD cycles with 60 (sample A), 40 (sample B), 30 (sample C), and 20 (sample D) $N_2^*$-based Pt ALD cycles. **a** ALD recipe scheme. For clarity, fewer than the real number of ALD cycles are schematically represented. **b** Experimental 2D GISAXS images. The dashed vertical lines indicate the $q_y$-position of the main scattering lobe. **c** Schematic representations of the GISAXS results. The Pt loading, mean center-to-center particle distance $\langle D \rangle$, average particle width $\langle W \rangle$, and average particle height $\langle H \rangle$ are indicated. **d** SEM images with 50 nm scale bars

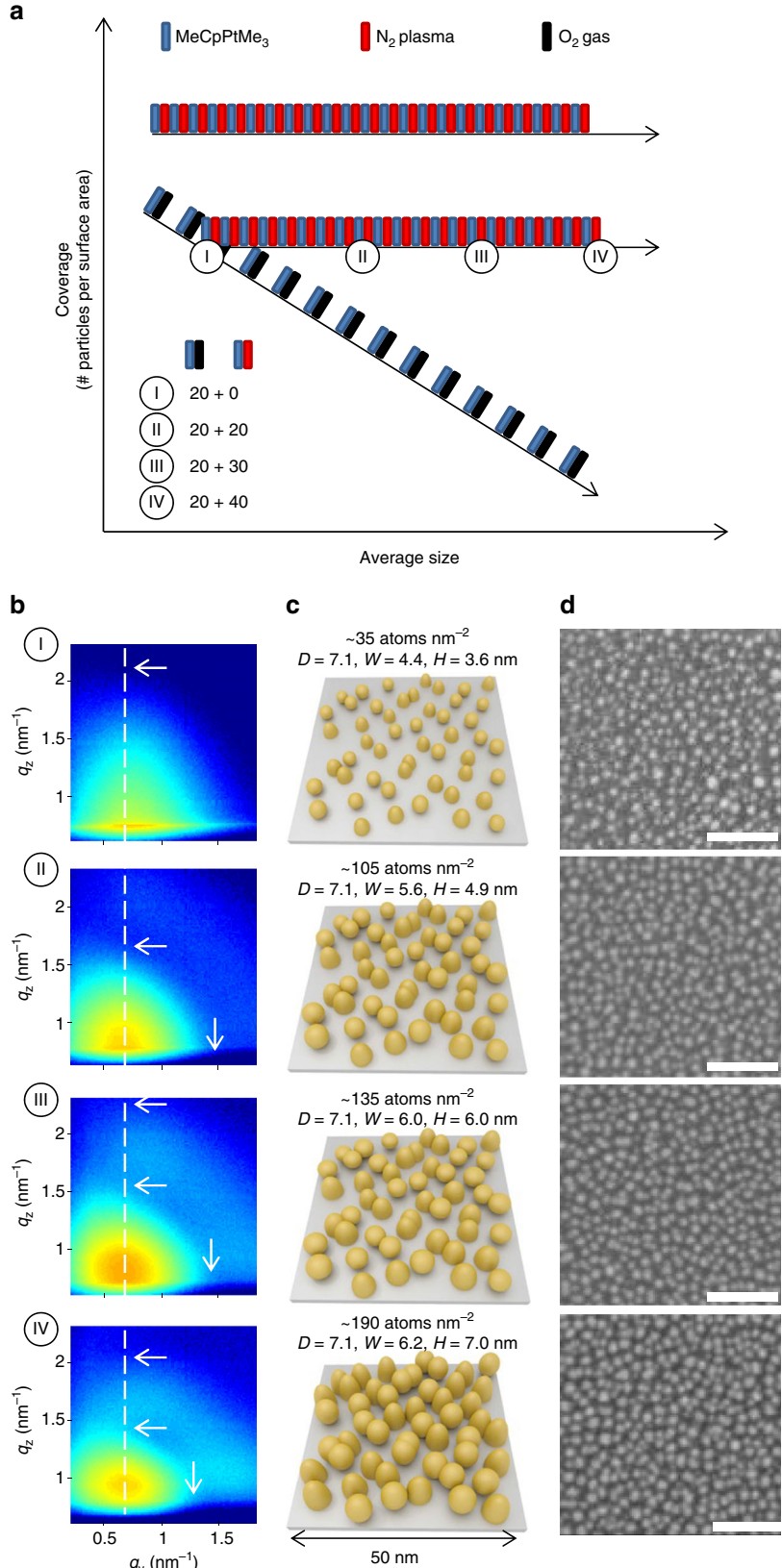

**Fig. 7** Precise control of size of Pt nanoparticles using ALD. Tuning the particle size at a fixed particle coverage by combining 20 $O_2$-based Pt ALD cycles with 0 (sample I), 20 (sample II), 30 (sample III), and 40 (sample IV) $N_2^*$-based Pt ALD cycles. **a** ALD recipe scheme. For clarity, fewer than the real number of ALD cycles are schematically represented. **b** Experimental 2D GISAXS images. The dashed vertical lines indicate the $q_y$-position of the main scattering lobe. The horizontal/vertical arrows indicate the minima along the $q_z/q_y$-direction. **c** Schematic representations of the GISAXS results. The Pt loading, mean center-to-center particle distance ⟨$D$⟩, average particle width ⟨$W$⟩, and average particle height ⟨$H$⟩ are indicated. **d** SEM images with 50 nm scale bars

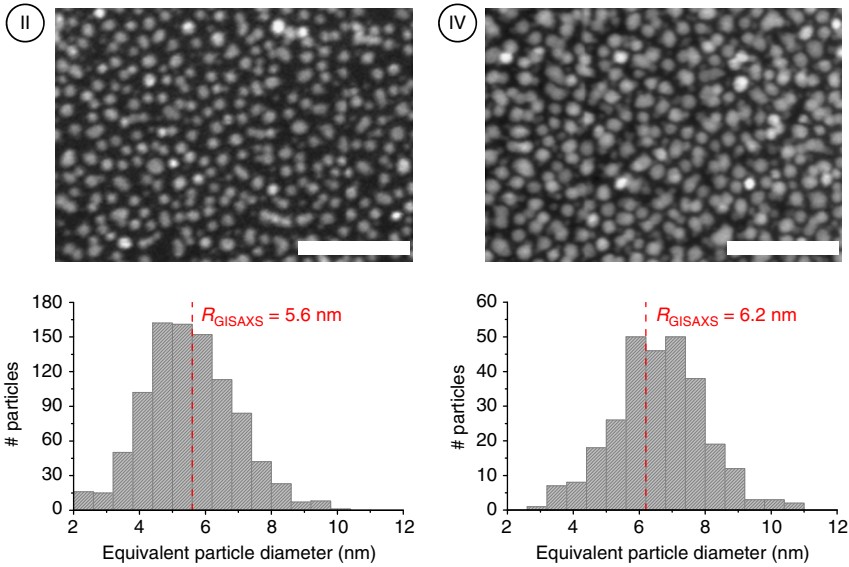

**Fig. 8** HAADF-STEM characterization of the Pt nanoparticle size distribution. Tuning the particle size by combining 20 $O_2$-based Pt ALD cycles with 20 (sample II) and 40 (sample IV) $N_2^*$-based Pt ALD cycles. HAADF-STEM images with 50 nm scale bars and derived particle size distributions

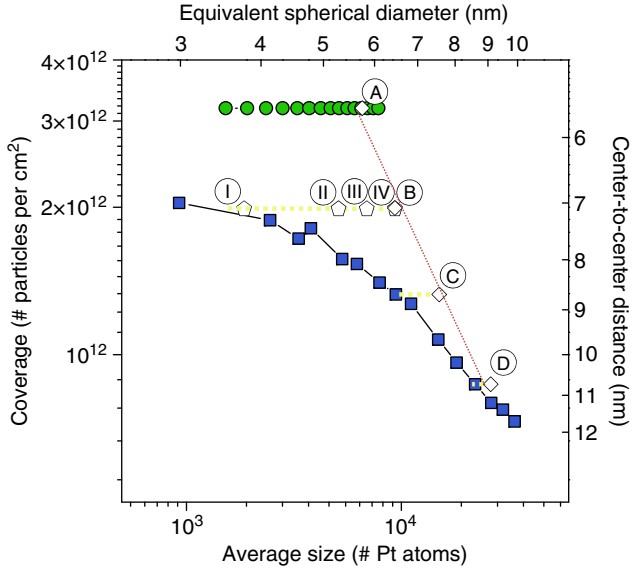

**Fig. 9** Precise control of coverage and size of Pt nanoparticles using ALD. Average particle coverage against particle size for samples A, B, C, and D (see also Fig. 6) and I, II, III, and IV (see also Fig. 7), together with the in situ measured data for the $O_2$-based (blue squares) and $N_2^*$-based (green circles) Pt ALD processes. The dashed yellow and red lines serve as guides to the eye

analysis of the GISAXS patterns are indicated and schematically represented in Fig. 7c (GISAXS analysis details and simulated patterns may be found in Supplementary Note 2 and Supplementary Fig. 11). The values for the average particle width obtained for samples II and IV are in good agreement with particle size distributions derived from high angle annular dark field (HAADF) scanning transmission electron microscopy (STEM) images (Fig. 8). The combined GISAXS and electron microscopy analysis thus confirms the capability of ALD to tailor the average Pt particle size with sub-nanometer precision while keeping the particle areal density constant.

Finally, the simulation results for samples A, B, C, and D and I, II, III, and IV are plotted in a coverage vs. size diagram together

with the in situ determined reference growth curves for the $O_2$ and $N_2^*$-based Pt ALD processes (Fig. 9). The coverage and size values obtained from GISAXS are in line with the expected ones. It is clear that the A, B, C, D sample set resulted in the expected coverage tuning, while the I, II, III, IV sample set resulted in a tuning of the particle size at fixed particle coverage. This figure demonstrates the high level of controllability of the ALD-based approach.

**Electrocatalytic characterization.** To demonstrate that the choice of reactant in Pt ALD is a determining parameter not only for the morphology of the deposited nanoparticles but also for their catalytic activity, Pt nanoparticles deposited via the $O_2$-based and $N_2^*$-based Pt ALD processes are evaluated in the hydrogen evolution reaction (HER), which is one half reaction in water electrolysis. Platinum is known to be the most active catalyst for the HER in acidic media[4, 5]. Dispersing the Pt into nanoparticles, so as to maximize the surface over volume ratio, is a common way to enhance activity and to reduce the Pt content and cost[4, 5, 59–61]. Fluorine-doped tin oxide (FTO) coated glass is selected as a conductive oxide support for the fabrication of Pt ALD electrodes. Electrodes with low Pt loading (3.5 μg cm$^{-2}$) are prepared by tuning the number of ALD cycles of both the $O_2$-based and $N_2^*$-based process. The loading corresponds to ca. 107 Pt atoms per nm$^2$ of glass substrate; note, the Pt loading per nm$^2$ of FTO surface will be lower due to surface roughness. The morphology of Pt particles on FTO according to SEM (Supplementary Fig. 14) resembles well the earlier observations made for $O_2$-based Pt ALD on the Si/SiO$_2$ surface (Fig. 5d), suggesting a similar diffusion-mediated particle growth mechanism for the $O_2$-based process on FTO.

The electrochemical characterizations are performed in 0.5 M $H_2SO_4$ using a three-electrode cell. The activity of the catalysts is determined via cyclic voltammetry at a scan rate of 2 mV s$^{-1}$ (Supplementary Fig. 15). In the scanned potential region, the bare FTO-coated glass is inert. Both the $O_2$-based and $N_2^*$-based Pt ALD electrodes show HER activity, with a slightly better performance for the $N_2^*$-based sample. At an overpotential of 50 mV, the $N_2^*$-based sample and the $O_2$-based sample exhibit a current density of 13.3 and 11.5 mA cm$^{-2}$, respectively. The mass activities are comparable to other ALD-prepared Pt

electrocatalysts, despite higher mass loadings[4]. Our results show that mastering Pt particle size by ALD offers the potential for fine-tuning state-of-the-art HER catalysts. The choice of reactant in the Pt ALD process influences the electrochemical activity, most likely due to a difference in nanoparticle morphology, i.e., nanoparticle shape, size, and coverage.

## Discussion

In summary, ALD is a versatile method for synthesizing supported Pt nanoparticles. The in situ GISAXS and XRF data presented here reveal a different island growth mechanism for $O_2$-based vs. $N_2^*$-based ALD of Pt. While $O_2$ induces atom or cluster mobility on the surface and promotes ripening of the Pt nanoparticles during ALD growth, surface diffusion phenomena seem to be suppressed during $N_2^*$-based ALD. Consequently, increasing the number of $O_2$-based ALD cycles results not only in an increase of the particle size but also in a decrease of the particle areal density due to dynamic particle coalescence. In contrast, during $N_2^*$-based ALD of Pt, the particle areal density remains constant. Therefore, the choice of reactant in the ALD process may influence the catalytic properties of the deposited nanoparticles, as exemplified here for the HER.

The observed difference in particle growth mode inspired us to develop an ALD-based strategy for tailoring the nanoparticle morphology on the atomic scale with independent control over size and coverage. In a two-step Pt ALD process, $O_2$ gas is first used as reactant to reach a desired particle coverage with minimum particle size. Subsequently, $N_2^*$-based Pt ALD can be applied to increase the particle size, while keeping the particle coverage fixed. GISAXS is shown to be a powerful characterization tool in determining the tuning range and precision that can be achieved with this approach. Our work provides an effective synthesis route for the preparation of model systems aiming for an in-depth understanding of the relation between the Pt particle morphology and the materials' properties and performance.

A final concluding remark is that the insights presented here for the particle growth during the $O_2$-based ALD process for Pt may be equally relevant to other noble metal ALD processes using $O_2$ as a reactant for temperatures above 200 °C (e.g., for Ru, Os, Rh, Ir)[62]. This work might therefore motivate future experimental studies exploring the influence of $O_2$ on the surface atom and cluster mobility during noble metal ALD.

## Methods

**Sample preparation**. ALD was performed in a home-built vacuum system with a base pressure of $10^{-6}$ mbar. Pt ALD was conducted using MeCpPtMe$_3$ (Strem Chemicals, 99%) and either $O_2$ or $N_2^*$. The Pt precursor was contained in a stainless steel container heated to 30 °C and Ar was used as a carrier gas. The precursor exposure time was 15 s, the precursor pumping time 30 s, the $O_2/N_2^*$ exposure time 10 s and the $O_2/N_2^*$ pumping time 15 s. During the precursor exposure, the pressure in the chamber was ca. 1 mbar, during the $O_2$ exposure ca. 1 mbar and during the $N_2^*$ exposure ca. $10^{-2}$ mbar. For the precursor and $O_2$, a static mode was applied, meaning that the valves to the pumping system were closed during the exposures.

**In situ characterization during ALD**. XRF and GISAXS measurements were performed during ALD at the SixS beamline of the SOLEIL synchrotron. Every 2 ALD cycles, the last pumping step was prolonged to allow for irradiation of the sample with 12.75 keV X-rays and detecting the fluorescent and scattered X-rays. The incidence angle was set to 1.2° for XRF and 0.5° for GISAXS. The acquisition times were 30 s for XRF and 20 s for GISAXS. The fluorescence spectra were recorded with a Röntek energy-dispersive silicon drift detector. The 2D GISAXS patterns were acquired with a MARccd detector positioned at a distance of approximately 1.7 m from the sample. In case of $N_2^*$-based Pt ALD, an additional $N_2^*$ exposure step was introduced following the data acquisition and preceding the next precursor pulse, as our previous work showed a drastic decrease in growth rate with increased pumping time between the plasma step and the precursor pulse[37].

**Ex situ morphological characterization**. GISAXS measurements were performed at the DUBBLE BM26B beamline of the ESRF synchrotron. The samples were irradiated with 12 keV X-rays at an incidence angle of 0.5°. The scattered X-rays were recorded with a Dectris Pilatus 1 M detector positioned at a distance of ~4.1 m from the sample. The acquisition time was 60 s. Scanning electron microscopy (SEM) was performed using a field emission gun SEM (Sirion, FEI) equipped with a through-the-lens secondary electron detector (TLD-SE). Plan-view samples for transmission electron microscopy (TEM) characterization were prepared by polishing the uncoated sides of silicon wafers using mechanical grinding and consequent thinning in a precision ion polishing system (Gatan Duo Mill 600). High-angle annular dark field scanning TEM (HAADF-STEM) images were collected using an aberration-corrected cubed FEI Titan operated at 300 kV.

**Ex situ elemental characterization**. The Pt loading was determined with a Bruker Artax XRF system with Mo X-ray source. Rutherford backscattering spectrometry (RBS) was used to calibrate the XRF spectrometer by determining the Pt loading of a representative series of four Pt on Si samples. A linear relation was found between the Pt loading from RBS and the Pt Lα XRF intensity measured for the same samples. In RBS, a 1.57 MeV He+ beam bombarded the sample at a tilt angle of 6° and a passivated implanted planar silicon (PIPS) detector was set at 14.8° with respect to the incident beam to collect the spectrum.

**Electrocatalytic testing**. Experimental details are available in the Supplementary Methods.

**Data availability**. The data that support the findings of this study are available from the corresponding author upon request.

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

## Acknowledgements

This research was funded by the Research Foundation–Flanders (FWO), the Special Research Fund BOF of Ghent University (GOA 01G01513) and the Flemish Government (Medium-scale research infrastructure funding-Hercules funding). J.D., T.D. and M.M.M. acknowledge the FWO for a research fellowship. S.B. acknowledges the European Research Council, ERC grant no. 335078–Colouratom. For the GISAXS and XRF measurements at SOLEIL, the authors received funding from the European Community's Trans National Access Program CALIPSO. We are also grateful to the SOLEIL and ESRF staff for smoothly running the facilities. The authors thank G. Verellen for his help with drawing the 3D sketches.

## Author contributions

J.D. designed the experiments, performed XRF and GISAXS measurements and analyzed the results; R.K.R. and E.S. assisted in preparing samples and performing XRF and GISAXS measurements; M.M.M., T.D., K.D.-C., G.P. and A.C. assisted in performing XRF and GISAXS measurements; M.K. and S.B. performed HAADF-STEM measurements and electron tomography; L.G., G.H., J.R. and J.A.M. performed electrocatalysis

measurements, A.V. performed RBS measurements; C.D. assisted in designing the experiments and analyzing the results. J.D. prepared the manuscript. All the authors contributed to revising the manuscript.

## Additional information

**Competing interests:** The authors declare no competing financial interests.

