## [Peer Review File · Nature Communications]

Reviewers' comments:

Reviewer #1 (Remarks to the Author):

The authors present an impressive study about the Pt nanoparticle growth during ALD. The use of in situ GISAXS and XRF give deep insights into the structures with a level of precision missing so far. However, several points require attention to render this work suitable for publication:

- 1) The authors should give better account to the already existing in situ studies on metal deposition using GISAXS by the Roth and Muller-Buschbaum groups. There are significantly more articles published which (in part) should be included into the introduction rather than be hidden in the supporting information. Since the existing work mainly is devoted to metals such as Au, Al and Ag and use sputter deposition, citation of these articles will not harm the publication of the present study.
- 2) The simulations are hidden in the supporting information. As the simulations are the backbone of the analysis I strongly suggest to include them in the main part with plotting the 2d simulated patterns, e.g. add simulations to figure 1 (and others shown the 2d GISAXS data). This will make the work significantly more convincing and easier to read.
- 3) In the support information more details to the simulations need to be added. In IsGISAXS there are very many parameters which need to be set, which are not mentioned at all in the SI. I expect the authors to provide them in the revised version, since else it is impossible to judge the used simulations properly! Perhaps use of a table might be appropriate to show the parameters.
- 4) The authors assume a particle shape. As shown by Schwartzkopf this can be proven in simulations. The authors are requested to provide such simulations as well to justify the choice of particle shape. Actually, the real particle shape is under massive debate in many cases and GISAXS is a unique tool to provide such information. The authors need also to consider that particle shape might change with loading.
- 5) Actually, the used mixed particle shapes of full and hemi spheroids make me wonder and are unexpected. Justification will be necessary beyond best agreement with the data, since such wetting conditions are very unexpected. Somehow I do not see this in the real space realizations in the main manuscript.
- 6) Error bars should be shown (e.g. figures 2a and 2d). Also the used size distributions should be elucidated.
- 7) The authors use a 2D paracrystal model, which is very unfortunate because it is not existing as proven by Wilhem Ruland in Makromol. Chem. 177, 3601-3617 (1976). Only a 1D paracrystal model is meaningful. Will this make any problem in the simulations?
- 8) The differences between data and simulation in the low q_y region is NOT due to a beamstop problem but due to the used modeling. It means that in the model large scattering objects are existing which cannot be found in the real samples. They might be caused from the tail in the size distribution function and likely can be eliminated by truncating them. I do not consider this as a very big problem, but the authors should at minimum give a proper explanation instead of the beamstop story.

Reviewer #2 (Remarks to the Author):

The manuscript reports on that what is exactly expressed by the title: The independent tuning of size and coverage of supported Pt nanoparticles using ALD. In principle this is an interesting feature that builds on extensive other work that has been published in this area:

- The preparation of supported nanoparticles by ALD for catalysis applications
- The precise size-control of these nanoparticles (also for particles smaller than reported here)
- The preparation of nanoparticles consisting of several materials (Pt, Ru, Pd), also in mixed phases (alloys) or in core/shell configurations
- The preparation of nanoparticles on highly structured materials (e.g. on nanosphere supports)
- The protection of the nanoparticles by overcoatings - The area-selective ALD of nanoparticles
- The demonstration of the activity of ALD-prepared nanoparticles in several heterogenous catalysis reactions (viz. various dehydrogenation and oxidation reactions)
- Etc.

This work goes back to the basis of the field and the (only) novelty of the work is that the authors show that the use of a N_2^* plasma as reactant allows for increasing the size of the Pt nanoparticles without changing the coverage of the Pt nanoparticles in terms of the number of particles per surface area. With the common chemistry using O_2 as the reactant, the size can be controlled but not without affecting the coverage of the nanoparticles.

As mentioned, this is a nice feature but by itself it does not warrant the publication of the work in Nature Communications. More important achievements (see above) have already been reported in high impact journals (including several Nature and Science journals). Furthermore, it is not clear what the impact of this work exactly is. The size of the Pt nanoparticles synthesized is relatively large (for optimized catalytic reactivity) whereas the 2-step method does not allow to increase the coverage of the nanoparticles over coverages obtained by the O_2 -based chemistry. The method might only be viable for Pt and not for the preparation of Ru and Pd nanoparticles and there alloys (with Pt). It is also questionable whether the N_2^* plasma approach allows for the preparation of the nanoparticles on highly structured materials (plasma cannot penetrate such materials). Moreover, it is not clear how the results rely on the specific reactor conditions employed by the authors. I can imagine that the N_2^* plasma conditions are very system-dependent and it is not obvious that similar results can be obtained by others. It might also only work SiO_2 supports and not on other support materials. Finally, another vital point, and perhaps the most important one, is that the authors have not demonstrated the catalytic activity of the nanoparticles at all. Considering the existing literature, I think that this should be a requirement for publication of the results in a high-impact journal.

To summarize, I don't question the novelty of the claim of this manuscript but I do question whether the impact of the claim is really demonstrated. The manuscript will only have sufficient impact and be of wide interest to the community at large if the improved catalytic performance of nanoparticles prepared by this two-step method is really demonstrated.

Finally, there is another point I would like to raise: to my opinion, the introduction seems to be biased. I don't think it is sufficiently comprehensive and it does not acknowledge the major achievements within the field.

Reviewer #3 (Remarks to the Author):

The manuscript is written well. It proposes a strategy to allow independent control of Pt particle size and coverage for nano-sized supported particles using a combination of O₂-based and N₂ plasma-based atomic layer deposition (ALD). Using the ALD method to synthesize model Pt nanoparticles has a lot of advantages compared to the use of more classical methods. The ALD method is a precise deposition techniques with good control over e.g. conformality, thickness, and composition. The field

To the reviewers best knowledge the authors report independent tuning of size and coverage of Pt nanoparticles for the the first time. This is not only of great interest in the field of heterogeneous catalysis but also e.g. in surface science and micro-electronics. It could be speculated that a similar strategy can be used to deposit other (combinations of) metals.

The manuscript shows the independent control of size and coverage using a variety of techniques (GISAXS, XRD, SEM, HAADF-STEM) appropriate to characterize Pt particles. The data presented in the manuscript supports the conclusions well.

Putting the details of the GISAXS simulations in the supplementary information is a good decision. It would be beneficial to corroborate some of the assumptions in the GISAXS analysis. E.g. the validity of the assumption of a log-normal distribution function (line 46 p.3, Suppl. Inf.) and a relative width of $\sigma = 1.1$. The GISAXS simulations show good qualitative agreement with the experimental data, however a few cross-sections through the 2-dimensional data for e.g. constant q_y or q_z could also show a quantitative comparison (for figures S2, S3, S4, S5, S6). In this way the validity of the implicit choice of fixing the size distribution and fitting the shape (in stead of the other way around) could be shown. As GISAXS is the main technique used to extract relevant parameters from the experiments, a quantitative comparison between simulations and experiments will strengthen the authors conclusions. As all 2 dimensional data is available it will not generate much extra work for the authors to show a quantitative comparison.

A similar quantitative analysis of the SEM results (in stead of the more limited qualitative

analysis shown on page 15, line 256 and in figures 5b and 6b, will make the claims even more convincing.

Therefore, I recommend publication with minor additions to the data analysis of GISAXS and SEM.

Responses to the reviewers' comments about the manuscript.

Reviewers' comments:

Reviewer #1 (Remarks to the Author):

The authors present an impressive study about the Pt nanoparticle growth during ALD. The use of in situ GISAXS and XRF give deep insights into the structures with a level of precision missing so far.

Author reply:

We thank the reviewer for the careful evaluation of our work and for the useful suggestions to further improve the manuscript. The manuscript has been modified according to the specific comments of the reviewer, as explained below.

However, several points require attention to render this work suitable for publication:

1) The authors should give better account to the already existing in situ studies on metal deposition using GISAXS by the Roth and Muller-Buschbaum groups. There are significantly more articles published which (in part) should be included into the introduction rather than be hidden in the supporting information. Since the existing work mainly is devoted to metals such as Au, Al and Ag and use sputter deposition, citation of these articles will not harm the publication of the present study.

Author reply:

As suggested by the reviewer, the introduction of the paper has been extended with a paragraph discussing prior work on *in situ* GISAXS studies during deposition processes in high vacuum, including the studies on metal sputtering by the Roth and Muller-Buschbaum groups.

Changes to the manuscript:

The following sentences have been added to the introduction: "Grazing incidence small angle x-ray scattering (GISAXS) is a synchrotron-based morphological characterization technique⁴¹ that requires no special sample preparation and is therefore ideally suited for a variety of in situ experiments, including the characterization of deposition processes in high vacuum.⁴² The technique has, for example, been used for real-time monitoring of noble metal growth by evaporation⁴³ and sputtering.⁴⁴⁻⁴⁷".

The following references have been added to the manuscript: "(41) Renaud, G., Lazzari, R. & Leroy, F. Probing surface and interface morphology with grazing incidence small angle x-ray scattering Surf. Sci. Rep. 64, 255-380 (2009). (42) Renaud, G., Lazzari, R., Revenant, C., Barbier, A., Noblet, M., Ulrich, O., Leroy, F., Jupille, J., Borensztein, Y., Henry, C. R., Deville, J.-P., Scheurer, F., Mane-Mane, J. & Fruchart, O. Science 300 1416-1419 (2003). (43) Lazzari, R., Leroy, F. & Renaud, G. Grazing-incidence small-angle x-ray scattering from dense packing of islands on surfaces: development of distorted wave Born approximation and correlation between particle sizes and spacing. Phys. Rev. B 76, 125411 (2007). (45) Santoro, G., Yu, S., Schwartzkopf, M., Zhang, P., Sarathlal, K. V., Risch, J. F. H., Rübhausen, M. A.,

Hernández, M., Domingo, C. & Roth, S. V. Silver substrates for surface enhanced Raman scattering: correlation between nanostructure and Raman scattering enhancement. *Appl. Phys. Lett.* **104**, 243107 (2014). (46) Schwartzkopf, M., Santoro, G., Brett, C. J., Rothkirch, A., Polonskyi, O., Hinz, A., Metwalli, E., Yao, Y., Strunskus, T., Faupel, F., Müller-Buschbaum, P. & Roth, S. V. Real-time monitoring of morphology and optical properties during sputter deposition for tailoring metal–polymer interfaces. *ACS Appl. Mater. Interfaces* **7**, 13547–13556 (2015). (47) Schwartzkopf, M. & Roth, S. V. Investigating polymer–metal interfaces by grazing incidence small-angle x-ray scattering from gradients to real-time studies. *Nanomaterials* **6**, 239 (2016).”

2) The simulations are hidden in the supporting information. As the simulations are the backbone of the analysis I strongly suggest to include them in the main part with plotting the 2d simulated patterns, e.g. add simulations to figure 1 (and others shown the 2d GISAXS data). This will make the work significantly more convincing and easier to read.

Author reply:

This advice is somewhat contradictory to the opinion of Reviewer #3 who stated “Putting the details of the GISAXS simulations in the supplementary information is a good decision”. Although the GISAXS analysis method and simulations are a very important part of the reported research, we believe that including it in the main text would distract the readers’ attention from the main scientific message of the manuscript, being the proposed ALD-based tuning strategy. Therefore, the GISAXS details and simulations are still included in the Supplementary Information. However, to give the reader an impression of the simulations results, the 2D simulated patterns have also been included next to the experimental GISAXS images in Figure 1 (but not for the other figures showing experimental 2D data). The reader is referred to the Supplementary Information for an overview of the input parameters and assumptions used for the simulations.

Changes to the manuscript:

The simulated 2D patterns have been added to Figure 1 in the main text.

The following sentences have been added to the main text: “Inspired by the work of Schwartzkopf et al.⁴⁴, a GISAXS analysis strategy is developed to obtain cycle-per-cycle information about the average particle distance and dimensions. The analysis approach is extensively explained in the Supplementary Information. The obtained morphological parameters for the O₂-based and N₂^{*}-based Pt ALD processes are compared in Figure 3a. To validate the analysis strategy, the experimental GISAXS images are compared with simulated patterns obtained with the software IsGISAXS.⁴⁹ These simulations assume a spheroidal shape for the Pt nanoparticles and use, amongst other parameters, the average center-to-center distance, particle width and particle height obtained from the analysis as input (all details can be found in the Supplementary Information). **Figure 1b shows the simulations that correspond to the experimental data in Figure 1a.** As indicated by the white arrows and dashed lines, the positions of the different maxima/minima are successfully reproduced, justifying our analysis strategy to derive the average morphological parameters and confirming the spheroidal shape of the nanoparticles.”

3) In the support information more details to the simulations need to be added. In IsGISAXS there are very many parameters which need to be set, which are not mentioned at all in the SI. I expect the authors to provide them in the revised version, since else it is impossible to judge the used simulations properly! Perhaps use of a table might be appropriate to show the parameters.

Author reply and changes to the manuscript:

We thank the reviewer for this advice. The description of our GISAXS analysis approach has been largely extended in the revised version of the Supplementary Information. Several Supplementary Figures (2, 3, 4, 5 and 6) have been added to clarify our analysis approach to the reader. As suggested by the reviewer, a table is added which includes the main input parameters for the IsGISAXS software.

4) The authors assume a particle shape. As shown by Schwartzkopf this can be proven in simulations. The authors are requested to provide such simulations as well to justify the choice of particle shape. Actually, the real particle shape is under massive debate in many cases and GISAXS is a unique tool to provide such information. The authors need also to consider that particle shape might change with loading.

Author reply:

We agree with the reviewer that Schwartzkopf et al. showed convincing proof for the ability of GISAXS to deduce the contact angle of truncated spheroidal particles in a study of a gold sputtering process [Nanoscale 5, 5053, 2013]. Based on simulations they showed a clear relation between the spheroid contact angle and the position of the second order of the first height peak in the scattering pattern, allowing to extract the nanoparticle contact angle based on the position of this second order scattering feature. However, in the recorded patterns for the Pt ALD processes, no such second order scattering feature can be observed. This suggests that the contact angle of the particles differs from particle to particle, causing smoothening of this second order scattering feature.

As requested by the reviewer, we have added Supplementary Figure 6 that compares 2D patterns simulated for different spheroidal Pt nanoparticles. The best agreement with the experiment was found when a mixture of full spheroids and hemispheroids is used for form factor calculation. We believe that this mixture of two different wetting conditions in the simulations corresponds to a real situation where the contact angle of the spheroidal Pt nanoparticles with the SiO₂ surface varies from particle to particle.

To further justify our choice of particle shape, we performed an electron tomography characterization of ALD-grown Pt nanoparticles (Figure 2 in the main text). This “3D TEM” imaging technique confirmed the spheroidal shape of the Pt nanoparticles, in agreement with GISAXS. Although the 3D tomogram of the Pt nanoparticles suggests that the contact angle of the Pt nanoparticles indeed varies from particle to particle, one should be careful with this kind of interpretation due to so-called missing wedge artifacts during the reconstruction of the TEM tomography.

Finally, we agree with the reviewer that the particle shape might change with Pt loading. We indeed noticed that the agreement between experiment and simulations is worse for Pt loadings that are higher than those reported in this work, i.e. when the Pt cluster morphology is dominated by worm-like and coalesced Pt structures, as revealed by SEM imaging. However, for all GISAXS patterns and Pt loadings considered in this work, the morphology is mostly dominated by isolated nanoparticles and in this case, the assumed two-particle model yields reasonable agreement with the experimental data.

Changes to the manuscript:

The following paragraph has been added to the Supplementary Information: “Finally, to motivate our two-particle model for calculating the form factor, Supplementary Figure 6 compares 2D patterns simulated for different spheroidal Pt nanoparticle geometries. In these simulations, the values for $\langle D \rangle$, $\langle H \rangle$ and $\langle W \rangle$, and those for ω and σ_R are kept constant, but the form factor is calculated for 100% full spheroids, 100% hemispheroids or a 50 to 50% mixture of both particle types. In the experimental 2D GISAXS pattern, one observes next to two clear scattering peaks a less intense arc-like feature (marked by 1 in Supplementary Figure 6) and a triangular scattering that emerges from the main scattering peak (marked by 2). Note that these scattering features are apparent in most of the experimental patterns recorded in this study. However, in the simulated scattering patterns for the one-particle models, one observes only one of these scattering features. In case of 100% full spheroids, the pattern is marked by a clear arc-like feature. In case of 100% hemispheroids, a clear scattering feature emerges from the main peak. By assuming a mixture of the two particle types for calculation of the form factor, the appearance of the two scattering features, as observed in the experimental patterns, can be reproduced in the simulations. We believe that this mixture of two different wetting conditions in the simulations corresponds to a real situation where the contact angle of the spheroidal Pt nanoparticles with the SiO₂ surface varies from particle to particle, as also suggested by the TEM tomography result.”

Supplementary Figure 6 | Effect of particle shape on GISAXS simulations. Comparison between experimental and simulated GISAXS patterns calculated for different spheroidal particle shapes. The particle shape assumed for calculation of the form factor is displayed in the top right corner of the respective simulated 2D GISAXS pattern. The table includes the input parameters that were used for the calculations.

The following paragraph has been added to the main text: “Analysis of the GISAXS data through comparison with simulations assumes a certain shape for the Pt nanoparticles. Therefore, **to obtain more insights in the 3D shape of the nanoparticles, an electron tomography study is performed** on a sample with a Pt loading of ~ 45 atoms / nm^2 prepared by the O_2 -based ALD process. While conventional transmission electron microscopy (TEM) only yields a 2D projection of a 3D object, electron tomography allows reconstructing the 3D structure of the object based on a large number of 2D projection images.⁴⁹ To acquire a full tilt range of 2D projection images with high angle annular dark field (HAADF) scanning transmission electron microscopy (STEM), a plan-view sample is prepared as explained in the Methods section and mounted on a dedicated tomography holder. After acquisition and alignment of the HAADF-STEM images, the “Simultaneous Iterative Reconstruction Technique” (SIRT) is used for the reconstruction of the 3D structure of the specimen. The reconstructed volume of Pt nanoparticles deposited in a ca. 60 by 60 nm^2 region on the Si/SiO_2 surface is visualized in Figure 2. An animated version of the tomogram is provided in the Supplementary Information as a video. The majority of the Pt clusters have a spheroidal shape, while some clusters consist of agglomerated smaller particles. Rounded particles are expected to expose many atomic steps and kinks presenting high catalytic activity.^{50,51} In the animated version of the tomogram, it can also be observed that some of the particles exhibit a flat surface at the Pt/SiO_2 interface (particles indicated by an arrow in Figure 2), while others seem to be full spheroids. The tomography study thus suggests that the contact angle of the Pt nanoparticles with the surface varies from particle to particle. However, it should be kept in mind that the tomography series was carried out on a plan-view TEM sample as indicated previously. Due to missing wedge artifacts,⁴⁹ which are more pronounced at the interface between the nanoparticles and the substrate in the case of plan”

Figure 2 | Electron tomography characterization of the Pt nanoparticle shape. 3D volume rendering, viewed from the top, of Pt nanoparticles synthesized with the O_2 -based ALD process (Pt loading ~ 45 atoms / nm^2). The arrows indicate particles that exhibit a flat surface at the Pt/SiO_2 interface. The scale bar indicates 10 nm.

The following sentences have been added to the main text: “As exemplified in the Supplementary Information, best agreement with the experimental GISAXS patterns is obtained when a two particle model is used to describe the spheroidal particles. A mixture of 50% (75%) full spheroids and 50% (25%) hemi-spheroids is assumed to simulate the patterns for the O_2 -(N_2^*)-based ALD process. This mixture of two different wetting conditions in the simulations suggests a real situation where the contact angle of the spheroidal Pt nanoparticles with the SiO_2 surface varies from particle to particle, as also suggested by the tomography result. Moreover, GISAXS indicates (on average) larger contact angles (larger dewetting) for the Pt nanoparticles deposited via the N_2^* -based ALD process.”

5) Actually, the used mixed particle shapes of full and hemi spheroids make me wonder and are unexpected. Justification will be necessary beyond best agreement with the data, since such wetting conditions are very unexpected. Somehow I do not see this in the real space realizations in the main manuscript.

Author reply:

The idea to combine two different particle shapes in the simulations was based on a previous publication by Kaune et al. who used a model consisting of parallelepiped and spheroid particle geometries to describe the cluster shape of gold nanoparticles [ACS Appl. Mater. Interf., 1, 353, 2009]. As mentioned above, we believe that the mixture of two different wetting conditions in the simulations corresponds to a real situation where the contact angle of the spheroidal Pt nanoparticles with the SiO_2 surface varies from particle to particle. This has been confirmed by an electron tomography study, though one should keep in mind that the morphology of the Pt nanoparticles near the SiO_2 interface might vary slightly from what is obtained in tomography due to missing edge artifacts.

The reviewer indicates that the varying contact angle was not incorporated in the real space realizations of the GISAXS analyses. We agree that this might cause some confusion and have therefore updated the real space sketches.

Changes to the manuscript:

The real space sketches in the main text have been updated and include now also the variation in contact angle over the different nanoparticles.

6) Error bars should be shown (e.g. figures 2a and 2d). Also the used size distributions should be elucidated.

Author reply:

The reviewer advises to add error bars to the average morphological parameters that were extracted from GISAXS. However, determining the error on these values is not straightforward. Therefore, this comment by the reviewer has been implemented in the Supplementary Information by adding a figure that shows how sensitive GISAXS is to small variations of the average particle height and width.

Supplementary Figure 4 gives the reader a visual impression of how accurate the average particle dimensions can be determined with our GISAXS analysis method.

Secondly, the reviewer advised us to elucidate the used size distribution. We apologize that this was not well explained before. We have added an explanation and figure to the Supplementary Information to make this clearer to the reader. Note that the average morphological parameters that are mentioned in the main text are determined without assuming a particle size distribution, as is explained in detail in the Supplementary Information. The distribution in particle sizes is only introduced to generate the simulated 2D patterns, which are compared to the experimental data in order to validate our GISAXS analysis strategy.

In addition, as advised by Reviewer #3, we have added a quantitative analysis of the SEM images shown in Figure 6. Lognormal functions have been fitted to the derived particle size distributions (Supplementary Figure 12) as well as to the particle size distribution obtained from TEM in Figure 8. It was found that electron microscopy yields a larger distribution width than evaluated from GISAXS, similar to what Sanchez et al. observed in their characterization of Au nanoparticles [Sci. Rep. 3, 3414, 2013]. They attributed this discrepancy to the different sampling conditions: SEM/TEM are limited to a very small region of the sample, while GISAXS averages out over a much larger sample area.

Changes to the manuscript:

The following paragraph has been added to the Supplementary Information: “To demonstrate how sensitive GISAXS is to changes in the parameters $\langle H \rangle$ and $\langle W \rangle$, Supplementary Figure 4 illustrates the effect of systematic Ångstrom-level changes on the simulated patterns. The simulations show that changes in the particle height of 2Å can easily be distinguished by their change in oscillation period in the vertical line profile ((a), right graph). Similarly, a 2Å deviation in the particle radius (4Å in particle width) is shown to have a noticeable effect on the horizontal line profile ((b), left graph). In both cases, also the corresponding horizontal/vertical line profile has changed, though to a lesser extent.”

Supplementary Figure 4 | Sensitivity of GISAXS to changes in the average particle sizes. The sensitivity of GISAXS to particle height (a) and particle width (b) variations: experimental (black data points) and calculated (green, red and blue curves) 1D horizontal (left graph, $q_z = 0.722 \text{ nm}^{-1}$) and vertical (right graph, $q_y = 0.59 \text{ nm}^{-1}$) line profiles. The table includes the input parameters that were used for the calculations. For form factor calculation, a mixture of 50% full spheroids and 50% hemispheroids was used.

The following sentences have been added to the Supplementary Information: “To improve the agreement between simulation and experiment, the model that was used in step 2 to calculate the 1D line profiles is extended to account for the distribution in particle sizes. For the sake of simplicity, the particle height and width distributions are chosen to be coupled, in the sense that a distribution of particle radii at constant height/radius ratio implies also a distribution of particle heights. A lognormal distribution is assumed for the particle radius R , based on precedence in the literature⁷⁻¹⁰:

$$p(R) = \frac{1}{\sqrt{2\pi} R \ln(\sigma_R)} \exp\left(-\frac{1}{2} \left(\frac{\ln(R/\langle R \rangle)}{\ln(\sigma_R)}\right)^2\right) \quad (2)$$

with σ_R the dimensionless geometric standard deviation. The size distribution is kept equal for both types of particles in the model (full spheroids and hemispheroids). The calculations furthermore use the local monodisperse approximation (LMA) formalism, which is commonly used for polydispersed systems.⁵ As an illustration, Supplementary Figure 5 compares simulations with and without size distribution for the same sample as in Supplementary Figures 3 and 4. The obvious effect of the size distribution is smoothening of the 1D line profiles, leading to an improved agreement with the experimental data for a σ_R -value of 1.1. Since the aim of the complete 2D simulations is to validate the derived values for $\langle D \rangle$, $\langle H \rangle$ and $\langle W \rangle$ rather than to derive the exact width of the particle size distribution, the σ_R -parameter was not treated as a fitting parameter but is kept constant to 1.1 for all simulations performed in this study.”

Supplementary Figure 5 | Effect of size dispersion on GISAXS simulations. Comparison between experimental and simulated GISAXS patterns calculated without and with coupled size distribution for

the particle width and height: (left) 2D GISAXS patterns, (right) experimental (black data points) and calculated (red curves) 1D horizontal (top graph, $q_z = 0.722 \text{ nm}^{-1}$) and vertical (bottom graph, $q_y = 0.59 \text{ nm}^{-1}$) line profiles. The particle radius distribution is displayed in the top right corner of the respective simulated 2D GISAXS pattern. The other input parameters for the calculations are the same as those for the calculations in Supplementary Figure 4. For form factor calculation, a mixture of 50% full spheroids and 50% hemispheroids was used.

The following paragraph has been added to the Supplementary Information: “As shown in Figure 8 in the main text, a good agreement is found between the average particle radius obtained from TEM analysis and the one derived from the GISAXS analysis. Supplementary Figure 12 below presents additional analysis results for the SEM images included in Figure 6 of the main text, confirming again the agreement in average particle radius obtained from real-space electron microscopy measurements and reciprocal space GISAXS data. The black lines for samples A, B and C are fitted lognormal functions to the particle size distributions. The wide distribution observed for sample D is a consequence of the formation of wormlike structures when a large number of O_2 -based ALD cycles is applied. For all lognormal fits, the value for the dimensionless geometric standard deviation σ_R is ~ 1.30 . Similar fits to the size distributions obtained from TEM (Figure 8) yield a σ_R -value of ~ 1.25 . Both of these values are larger than the value of 1.1 evaluated from GISAXS. However, for GISAXS simulations with a σ_R -value of 1.25 or 1.30, the scattering features are highly smoothed or damped, in disagreement with the experimental patterns. Similar differences in particle radius distribution obtained from TEM and GISAXS have been observed before for 1-10 nm Au nanoparticles embedded in a SiO_2 film and may be attributed to different sampling conditions.¹² For our SEM and TEM analyses, 300 to 1000 particles are measured from a small region of the sample ($< 500 \times 500 \text{ nm}^2$) while GISAXS probes a sample area of ca. $300 \text{ nm} \times 2 \text{ cm}$, averaging over an estimated 10^8 particles.”

Supplementary Figure 12 | SEM characterization of the Pt nanoparticle size distribution. Tuning the particle coverage by combining 0 (sample A), 20 (sample B), 30 (sample C) and 40 (sample D) O_2 -based Pt ALD cycles with 60 (sample A), 40 (sample B), 30 (sample C) and 20 (sample D) N_2^* -based Pt ALD cycles. SEM images with 100 nm scale bars and derived particle size distributions. The black lines are fitted lognormal functions to the data.

The following references have been added to the Supplementary Information: “(7) Blackman, J. A., Evans, B. L. & Maarroof, A. I. Phys. Rev. B 49, 13863-13871 (1994). (8) Söderlund, J., Kiss, L. B., Niklasson, G. A., Granqvist, C. G. Phys. Rev. Lett. 80, 2386-2388 (1998). (9) Kiss, L. B., Söderlund, J., Niklasson, G. A., Granqvist, C. G. Nanotechnology 10, 25-28 (1999). (10) Meshot, E. R., Verploegen, E., Bedewy, M., Tawfick, S., Woll, A. R., Green, K. S., Hromalik, M., Koerner, L. J., Philipp, H. T., Tate, M. W., Gruner, S. M. & Hart, A. J. ACS Nano 6, 5091-5101 (2012). (11) Qadri, M. U., Diaz Diaz, A. F., Cittadini, M., Martucci, A., Pujol, M. C., Ferré-Borrull, J., Llobet, E., Aguiló, M. & Díaz, F. Sensors 14, 11427-11443 (2014). (12) Sanchez, D. F., Marmitt, G., Marin, C., Baptista, D. L., Azevedo, G. M., Grande, P. L. & Fichtner, P. F. P. Sci. Rep. 3, 3414 (2013).”

7) The authors use a 2D paracrystal model, which is very unfortunate because it is not existing as proven by Wilhem Ruland in Makromol. Chem. 177, 3601-3617 (1976). Only a 1D paracrystal model is meaningful. Will this make any problem in the simulations?

Author reply and changes to the manuscript:

We thank the reviewer for this very valuable comment. In fact, all simulations were done using a 1D paracrystal model in IsGISAXS. We mistakenly mentioned 2D paracrystal model because we assume a regular 2D lattice in the first step of the GISAXS analysis. However, for all calculations in IsGISAXS, we have used the 1D paracrystal model, as is now mentioned correctly in the Supplementary Information.

8) The differences between data and simulation in the low q_y region is NOT due to a beamstop problem but due to the used modeling. It means that in the model large scattering objects are existing which cannot be found in the real samples. They might be caused from the tail in the size distribution function and likely can be eliminated by truncating them. I do not consider this as a very big problem, but the authors should at minimum give a proper explanation instead of the beamstop story.

Author reply:

We thank the reviewer for this comment. The comment on the beam stop effect was actually based on experimental observations where a slightly different position of our *in vacuo* beam stop did have some effect on the horizontal cut in the low q_y region. However, we agree that this might not be the main explanation for the discrepancy near $q_y = 0 \text{ nm}^{-1}$, and that the explanation offered by the reviewer is more appropriate.

Changes to the manuscript:

The following sentence has been removed from the Supplementary Information: “The discrepancy with the experimental images near $q_y = 0 \text{ nm}^{-1}$ could result from beam stop shadowing.”

The following sentence has been added to the Supplementary Information: “The discrepancy with the experimental images near $q_y = 0 \text{ nm}^{-1}$ arises from the interference function in the simulations showing a tail towards low q_y -values originating from larger, more widely spaced particles which are not present in the real samples.”

Reviewer #2 (Remarks to the Author):

The manuscript reports on that what is exactly expressed by the title: The independent tuning of size and coverage of supported Pt nanoparticles using ALD. In principle this is an interesting feature that builds on extensive other work that has been published in this area:

- The preparation of supported nanoparticles by ALD for catalysis applications
- The precise size-control of these nanoparticles (also for particles smaller than reported here)
- The preparation of nanoparticles consisting of several materials (Pt, Ru, Pd), also in mixed phases (alloys) or in core/shell configurations
- The preparation of nanoparticles on highly structured materials (e.g. on nanosphere supports)
- The protection of the nanoparticles by overcoatings
- The area-selective ALD of nanoparticles
- The demonstration of the activity of ALD-prepared nanoparticles in several heterogenous catalysis reactions (viz. various dehydrogenation and oxidation reactions)
- Etc.

This work goes back to the basis of the field and the (only) novelty of the work is that the authors show that the use of a N_2^* plasma as reactant allows for increasing the size of the Pt nanoparticles without changing the coverage of the Pt nanoparticles in terms of the number of particles per surface area. With the common chemistry using O_2 as the reactant, the size can be controlled but not without affecting the coverage of the nanoparticles.

As mentioned, this is a nice feature but by itself it does not warrant the publication of the work in Nature Communications. More important achievements (see above) have already been reported in high impact journals (including several Nature and Science journals). Furthermore, it is not clear what the impact of this work exactly is.

Author reply:

We thank the reviewer for the careful evaluation of our work. We regret that the reviewer is not convinced by the significance and novelty of our work.

We agree with the reviewer that 'ALD for catalysis' is a growing research field and that several important works have been published before:

- **the preparation and size-control of supported nanoparticles, e.g. citations 25, 28-33**
- **the preparation of alloyed and core/shell nanoparticles, e.g. citations 20-24**
- **the preparation of nanoparticles on high surface area supports, e.g. citations 28-33**
- **the protection of the nanoparticles by overcoatings, e.g. citations 26, 27**
- **the area-selective ALD of nanoparticles, e.g. citation 24**
- **the demonstration of the activity of ALD-prepared nanoparticles in several heterogenous catalysis reactions, e.g. citations 28-33**

To stress the novelty of our work, we would like to respectfully remark that this work presents

- an accurate tuning strategy to independently control the Pt nanoparticle size and coverage, even at high surface densities of nanoparticles for which precise control is often difficult to achieve due to easy merging and sintering of the Pt nanoparticles;
- the first application of the N_2^* -based Pt ALD process for the growth of Pt nanoparticles;
- the first *in situ* characterization of the evolution in morphology during ALD of Pt and of noble metals in general, yielding insights in Pt particle ALD growth with a level of detail missing so far;
- the first convincing experimental proof to date of the important role of atom and cluster surface diffusion during the commonly applied O_2 -based Pt ALD process;
- clear experimental evidence of the important role of the choice of reactant used in noble metal ALD.

The reviewer mentions that it is not clear what the impact of the work exactly is. We are convinced that the manuscript presents novel insights in nanoparticle growth by ALD that are highly important to the development of model systems for catalysis research (the first two points listed above). The tuning strategy that is presented in this work will be useful to create systems that allow to elucidate the effect of *particle proximity* on (electro)catalytic activity and selectivity. This has proven challenging by using conventional synthesis methods, such as incipient wetness or precipitation, because the effect of the particle size (distribution) cannot be scrutinized independently from the nanoparticle coverage, and in turn, the particle distance. To illustrate that the particle distance is indeed a determining parameter which receives attention in current catalysis research, we refer to the works by Nesselberger et al. [Nature Materials 12, 919 (2013)] and Mistry et al. [ACS Catalysis 6, 1075 (2016)]. The former study revealed that the edge-to-edge particle distance between Pt clusters decisively influences the electrochemical oxygen reduction reaction (ORR) activity, while the latter study demonstrated that the particle distance between Cu nanoparticles plays a defining role in product selectivity during electrocatalytic reduction of CO_2 . We are convinced that the proposed ALD method offers an important novel strategy to further deconvolute the effect of nanoparticle size and distance in (electro)catalytic reactions.

Moreover, controlling the growth of noble metal ALD processes is not only important to the field of catalysis, but also to the field of microelectronics, as also indicated by referee 3. This manuscript presents an in-depth *in situ* characterization of the nucleation-controlled growth mode of Pt ALD and provides novel fundamental insights (the latter two point listed above) which broaden our understanding of noble metal ALD, important for catalysis and microelectronics, and which will inspire other researchers who focus on experimental or modeling studies of the nucleation of noble metal ALD processes.

The size of the Pt nanoparticles synthesized is relatively large (for optimized catalytic reactivity) whereas the 2-step method does not allow to increase the coverage of the nanoparticles over coverages obtained by the O_2 -based chemistry.

Author reply:

We respectfully remark that the highest coverage is obtained with the N_2^* -based ALD process, and not by the O_2 -based process as mentioned by the reviewer. Nevertheless, the reviewer is right that the introduced two-step method does not lead to higher coverages. However, the claim of the manuscript is not that the two-step process yields the most optimal morphologies for catalysis, but that the tuning potential in itself provides opportunities for fundamental catalysis research, as motivated above.

The reviewer mentions that the size of the Pt nanoparticles is relatively large. The lower limit for the particle sizes shown in this work, ca. 3-4 nm in diameter, is determined by the limit for which analysis of the GISAXS patterns can be performed in a reliable way. For lower Pt loadings, and thus smaller Pt nanoparticle sizes, a broad scattering peak without side minima/maxima is observed in GISAXS which is difficult to analyze. However, qualitative comparison of the scattering data at low Pt loadings for the O_2 -based vs. N_2^* -based Pt ALD process reveals similar trends as reported in the manuscript for Pt loadings > 45 atoms / nm^2 . The figure below shows horizontal line profiles through the scattering data and it can be observed that the peak maximum appears at higher q_y -values for the N_2^* -based process (dashed lines, $q_{y,max} \approx 1.0 \text{ nm}^{-1}$) than for the O_2 -based process (solid lines, $q_{y,max} \approx 0.75 \text{ nm}^{-1}$). This qualitative comparison suggests that the N_2^* -based process results in smaller nanoparticles with a smaller particle-to-particle distance than the O_2 -based process, also at these low Pt loadings.

Figure: Comparison of horizontal 1D line profiles through the scattering data measured *in situ* for low Pt loadings. Dashed lines: N_2^* -based Pt ALD process. Solid lines: O_2 -based Pt ALD process.

This result shows that the choice of reactant in the Pt ALD process impacts the morphology of the Pt nuclei from the very start of the ALD process. However, as mentioned, these data have not been included in the manuscript due to the lack of quantitative results for the morphological parameters $\langle D \rangle$, $\langle H \rangle$ and $\langle W \rangle$. On the other hand, we believe that the range of particle sizes presented in this work, 4-10 nm, offers sufficient opportunities for catalysis applications, and more specifically for the synthesis of model systems towards fundamental catalysis research. In the above cited work by Nesselberger et al. [Nature Materials 12, 919 (2013)], the particle diameter used in their calculations was 4 nm. Mistry et al. [ACS Catalysis 6, 1075 (2016)] focused on nanoparticles with diameters in the range 1.5-7 nm.

The method might only be viable for Pt and not for the preparation of Ru and Pd nanoparticles and their alloys (with Pt).

Author reply:

As indicated by the reviewer, the presented tuning method will not be directly applicable to other noble metal ALD processes due to different chemistries. However, the manuscript does contain results that are important and relevant to the whole field of noble metal ALD. There are several noble metal ALD processes – for Ru, Os, Rh, and Ir – that use O₂ as a reactant at deposition temperatures above 200 °C. Researchers so far have not paid much attention to the impact of O₂ on the surface mobility, coalescence regime (dynamic vs. static) and evolution in nanoparticle coverage. However, as shown here for the Pt ALD process, O₂ might have a strong influence on the particle growth and one should be aware that tuning the size by changing the number of ALD cycles might come at the cost of a decrease in particle coverage. Therefore, the results presented here will trigger new investigations on the role of O₂ in other noble metal ALD processes.

In addition, we see opportunities that similar tuning strategies, based on changing the reactant, can be developed for other noble metal ALD processes. For example, for the deposition of Pd nanoparticles, the Pd(hfac)₂ precursor can be combined with formalin, H₂, H₂ plasma or H₂ plasma followed by O₂ plasma. Since the O₂ plasma step in this latter process might cause diffusion-mediated coalescence like observed here for the oxidative Pt ALD chemistry, this could offer opportunities for tuning particle size and coverage by combining processes using different reactants.

Changes to the manuscript:

The following sentences have been added to the Discussion section of the manuscript: “A final concluding remark is that the insights presented here for the particle growth during the O₂-based ALD process for Pt may be equally relevant to other noble metal ALD processes using O₂ as a reactant for temperatures above 200°C (e.g. for Ru, Os, Rh, Ir).⁶⁰ This work might therefore motivate future experimental studies exploring the influence of O₂ on the surface atom and cluster mobility during noble metal ALD.”

It is also questionable whether the N₂* plasma approach allows for the preparation of the nanoparticles on highly structured materials (plasma cannot penetrate such materials). Moreover, it is not clear how the results rely on the specific reactor conditions employed by the authors. I can imagine that the N₂* plasma conditions are very system-dependent and it is not obvious that similar results can be obtained by others.

Author reply:

We agree with the reviewer that coating mesoporous materials presents a higher challenge for plasma-enhanced compared to thermal ALD. On the other hand, we would like to emphasize that PE-ALD can be applied successfully on high surface area (non-porous) powder particles [Longrie et al. Surf. Coat. Technol. 183, 213 (2012); ACS Appl. Mater. Interfaces 6, 7316 (2014)]. Since the proposed

tuning strategy is mainly aimed at synthesizing model systems for fundamental research, it will be possible to select a support that suits the ALD process.

The plasma-enhanced ALD reactor designed for this work is a standard pump-type ALD reactor with a remote inductively coupled plasma source, as reported by Rossnagel et al. [J. Vac. Sci. Technol., B 18, 2016 (2000)] and commercially implemented by, e.g., Oxford Instruments and Cambridge Nanotech. All the details concerning the reactor are included in a recent publication which has been added to the reference list of the manuscript ((48) Dendooven, J., Solano, E., Minjauw, M. M., Van de Kerckhove, K., Coati, A., Fonda, E., Portale, G., Garreau, Y. & Detavernier, C. Mobile setup for synchrotron based *in situ* characterization during thermal and plasma-enhanced atomic layer deposition. Rev. Sci. Instrum. 87, 113905 (2016).). In addition, the N_2^* -based process has been performed in several reactors in our lab, including a rotary reactor for powder particle coating, and similar process characteristics have been obtained in all setups.

It might also only work SiO_2 supports and not on other support materials.

Author reply:

The reviewer is right that the interaction between the Pt nanoparticles and the underlying surface differs from support to support, influencing the particle nucleation and growth during ALD. However, this comment is valid for all ALD-based strategies related to nanoparticle growth. On the other hand, using our *in situ* methodology, we have monitored Pt ALD growth on different metal oxide surfaces and, although there are support-related effects, these results prove that the O_2 -based ALD process is governed by a diffusion-mediated growth, marked by a decrease in number of particles per surface area with increasing number of ALD cycles, on all tested supports (SiO_2 , TiO_2 , Al_2O_3). The figures below show the scattering data measured *in situ* during growth on TiO_2 and Al_2O_3 surfaces. The shift of the main scattering peak to lower q_y -values with increasing number of ALD cycles is indicative of an increasing center-to-center distance, and in turn, decreasing particle coverage. These results give a strong indication that the presented tuning strategy is extendable to other metal oxide surfaces.

Figure: Selection of experimental 2D GISAXS images measured in situ during O₂-based Pt ALD on TiO₂. The number in the top right corner is the ALD cycle number.

Figure: Selection of experimental 2D GISAXS images measured in situ during O₂-based Pt ALD on Al₂O₃. The number in the top right corner is the ALD cycle number.

Finally, another vital point, and perhaps the most important one, is that the authors have not demonstrated the catalytic activity of the nanoparticles at all. Considering the existing literature, I think that this should be a requirement for publication of the results in a high-impact journal.

Author reply:

As suggested by the reviewer, the catalytic activity of the Pt nanoparticles has been evaluated in a proof-of-principle experiment. Because nanoparticle sintering is an often encountered problem during catalytic reactions at elevated temperatures (which may be prevented with an oxide ALD overcoat) and the focus of this work is on the differences in as-deposited morphology with the choice of reactant in the ALD process, the activity of the Pt nanoparticles was tested in electrocatalysis. The electrochemical hydrogen evolution reaction (HER) was selected as probe reaction.

The electrocatalytic experiment required the deposition of Pt nanoparticles on a conductive support. As also discussed above, we have clear indications that the Pt nanoparticle growth behavior, as reported in the manuscript for a SiO₂ support, is similar on other metal oxide supports. Therefore, fluorine-doped tin oxide (FTO) coated glass was selected as a conductive oxide support. Low Pt-loading catalysts were prepared by depositing 3.5 μg Pt per cm² of FTO/glass substrate using both the O₂-based and N₂^{*}-based ALD processes. Both ALD processes resulted in Pt nanoparticles that are effective catalysts for water splitting, with a better performance for the N₂^{*}-based sample.

Changes to the manuscript:

The following paragraphs have been added to the main text: “To demonstrate that the choice of reactant in Pt ALD is a determining parameter not only for the morphology of the deposited nanoparticles but also for their catalytic activity, Pt nanoparticles deposited via the O₂-based and N₂^{*}-based Pt ALD processes are evaluated in the hydrogen evolution reaction (HER), which is one half reaction in water electrolysis. Platinum is known to be the most active catalyst for the HER in acidic media.^{4,5} Dispersing the Pt into nanoparticles, so as to maximize the surface over volume ratio, is a common way to enhance activity and to reduce the Pt content and cost.^{4,5,57-59} Fluorine-doped tin oxide (FTO) coated glass is selected as a conductive oxide support for the fabrication of Pt ALD electrodes. Electrodes with low Pt loading (3.5 μg/cm²) are prepared by tuning the number of ALD cycles of both the O₂-based and N₂^{*}-based process. The loading corresponds to ca. 107 Pt atoms / nm² of glass substrate; note, the Pt loading per nm² of FTO surface will be lower due to surface roughness. The morphology of Pt particles on FTO according to SEM (Supplementary Figure 13) resembles well the earlier observations made for O₂-based Pt ALD on the Si/SiO₂ surface (Figure 5(d)), suggesting a similar diffusion-mediated particle growth mechanism for the O₂-based process on FTO.

The electrochemical characterizations are performed in 0.5 M H₂SO₄ using a three-electrode cell. The activity of the catalysts is determined via cyclic voltammetry at a scan rate of 2 mV/s (Supplementary Figure 14). In the scanned potential region, the bare FTO coated glass is inert. Both the O₂-based and N₂^{*}-based Pt ALD electrodes show HER activity, with a slightly better performance for the N₂^{*}-based sample. At an overpotential of 50 mV, the N₂^{*}-based sample and the O₂-based sample exhibit a current density of 13.3 mA/cm² and 11.5 mA/cm², respectively. The mass activities are comparable to

other ALD-prepared Pt electrocatalysts, despite higher mass loadings.⁴ Our results show that mastering Pt particle size by ALD offers the potential for fine-tuning state-of-the-art HER catalysts. The choice of reactant in the Pt ALD process influences the electrochemical activity, most likely due to a difference in nanoparticle morphology, i.e. nanoparticle shape, size and coverage.”

The Supplementary Information has been updated with experimental details on the electrocatalytic testing and the following Supplementary Figures have been added:

Supplementary Figure 13 | Morphological characterization of Pt nanostructures on FTO. SEM images of worm-like Pt nanostructures deposited with the O₂-based Pt ALD process on a FTO coated glass slide. The sample contains ca. 5 μg of Pt per cm^2 of glass substrate. The scale bars indicate 500 nm (left) and 100 nm (right). In the right image, the contrast is enhanced for an improved visibility of the Pt nanostructures.

Supplementary Figure 14 | Evaluation of Pt nanoparticles in the HER of water electrolysis. Cyclic voltammograms of Pt nanoparticles deposited with the O₂-based and N₂*-based ALD processes on FTO coated glass slides. Scan rate was 2 mV/s. Both samples contain ca. 3.5 μg of Pt per cm² of glass substrate. The black curve was measured with a bare FTO-coated glass substrate. The current density is calculated based on the geometric surface area. Raw data were smoothed with a 20-point moving average. Forward and backward scans of a single cycle were averaged to obtain the plot. Turnover frequencies at 50 mV overpotential were calculated to be 4.1 s⁻¹ and 3.3 s⁻¹ for the N₂*- and O₂-based sample, respectively.

The following references have been added to the manuscript: “(4) Dasgupta, N. P., Liu, C., Andrews, S., Prinz, F. B. & Yang, P. Atomic layer deposition of platinum catalysts on nanowire surfaces for photoelectrochemical water reduction. *J. Am. Chem. Soc.* 135, 12932-12935 (2013). (5) Cheng, N., Stambula, S., Wang, D., Banis, M. N., Liu, J., Riese, A., Xiao, B., Li, R., Sham, T.-K., Liu, L.-M., Botton, G. A. & Sun, X. Platinum single-atom and cluster catalysis of the hydrogen evolution reaction. *Nat. Commun.* 7, 13638 (2016). (54) Domínguez-Crespo, M. A., Ramírez-Meneses, E., Torres-Huerta, A.M., Garibay-Febles, V. & Philippot, K. Kinetics of hydrogen evolution reaction on stabilized Ni, Pt and Ni-Pt nanoparticles obtained by an organometallic approach. *Int. J. Hydrogen Energy* 37, 4798-4811 (2012). (55) Tan, T. L., Wang, L.-L., Zhang, J., Johnson, D. D. & Bai, K. Platinum nanoparticle during electrochemical hydrogen evolution: adsorbate distribution, active reaction species, and size effect. *ACS Catal.* 5, 2376–2383 (2015). (56) Devadas, B. & Imae, T. Hydrogen evolution reaction efficiency by low loading of platinum nanoparticles protected by dendrimers on carbon materials. *Electrochem. Comm.* 72, 135-139 (2016).”

To summarize, I don't question the novelty of the claim of this manuscript but I do question whether the impact of the claim is really demonstrated. The manuscript will only have sufficient impact and be of wide interest to the community at large if the improved catalytic performance of nanoparticles prepared by this two-step method is really demonstrated.

Author reply:

As discussed above, the catalytic activity of the Pt nanoparticles synthesized with the O₂-based and N₂^{*}-based ALD processes is now demonstrated in the manuscript. However, we would like to clarify our opinion on the expected impact of the manuscript. What we demonstrate in the manuscript is that we can accurately control the Pt nanoparticle size and coverage by using different reactants in the Pt ALD process and applying the correct sequences. We do not claim in the manuscript that the two-step method will (necessarily) lead to improved catalytic performance. Instead, we envision that the tuning method will impact the field of fundamental catalysis research by offering a new way to synthesize precise model systems that allow to link catalytic (or catalysis-related) and morphological properties of the nanoparticles.

To support this statement, we show below a result of a recent study of some of the authors focusing on the coarsening behavior of Pt nanoparticles at elevated temperatures. The coarsening of nanoparticles is considered the main cause for thermal deactivation and lifetime reduction of supported catalysts. The tuning method proposed in this manuscript was used to synthesize samples with a well-controlled equal amount of Pt atoms per surface area but distinct as-deposited morphology. These samples were annealed in 18% O₂ in He while *in situ* GISAXS patterns were recorded. The figure below shows the evolution in nanoparticle radius with sample temperature and reveals that the onset temperature for particle coarsening increases with increasing particle size. This example illustrates that the tuning strategy presented in this manuscript can indeed be used to synthesize model systems enabling systematic characterizations of the effect of particle morphology on catalysis-related processes. We are convinced that many other examples will follow.

Figure: (left) Schemes depicting the as-deposited morphology of the different samples, as synthesized by the tuning method presented in this manuscript. (right) Particle radii determined from *in situ* GISAXS data measured during annealing (0.2 °C/s) of the samples under 18% O₂ in He. The colored arrows indicate the respective onset temperatures for coarsening.

Finally, there is another point I would like to raise: to my opinion, the introduction seems to be biased. I don't think it is sufficiently comprehensive and it does not acknowledge the major achievements within the field.

Author reply:

As suggested by the referee, the introduction of the paper has been extended with a paragraph discussing prior work in the field of noble metal ALD for catalysis.

Changes to the manuscript:

The following sentences have been added to the introduction: "This has led to several ALD-based strategies for the synthesis of monometallic and bimetallic nanoparticles.^{14,16,18} Well-chosen combinations of noble metal ALD chemistries and processing conditions can result in alloyed²⁰⁻²² or core/shell^{23,24} nanoparticles containing Pt, Pd and/or Ru. Another approach consisted of using metal oxide ALD prior to Pd deposition to modify the number of nucleation sites and hence nanoparticle loading.²⁵ Metal oxide ALD overcoats have also proven very effective to stabilize metal nanoparticles in high temperature reactions while preserving their catalytic activity.^{26,27}"

The following references have been added to the main manuscript: "(14) Lu, J. L., Elam, J. W. & Stair, P. C. Synthesis and stabilization of supported metal catalysts by atomic layer deposition. *Acc. Chem. Res.* 46, 1806-1815 (2013). (20) Christensen, S. T., Feng, H., Libera, J. L., Guo, N., Miller, J. T., Stair, P. C. & Elam, J. W. Supported Ru-Pt bimetallic nanoparticle catalysts prepared by atomic layer deposition. *Nano Lett.* 10, 3047-3051 (2010). (21) Lu, J. L., Low, K.-B., Lei, Y., Libera, J. A., Nicholls, A., Stair, P. C., & Elam, J. W. Toward atomically-precise synthesis of supported bimetallic nanoparticles using atomic layer deposition. *Nat. Commun.* 5, 3264 (2014). (22) Ramachandran, R. K., Dendooven, J., Filez, M., Galvita, V. V., Poelman, H., Solano, E., Minjauw, M. M., Devloo-Casier, K., Fonda, E., Hermida-Merino, D., Bras, W., Marin, G. B. & Detavernier, C. Atomic layer deposition route to tailor nanoalloys of noble and non-noble metals. *ACS Nano* 10, 8770-8777 (2016). (23) Weber, M. J., Mackus, A. J. M., Verheijen, M. A., van der Marel, C. & Kessels, W. M. M. Supported core/shell bimetallic nanoparticles synthesis by atomic layer deposition. *Chem. Mater.* 24, 2973-2977 (2012). (24) Cao, K., Zhu, Q., Shan, B. & Chen, R. Controlled synthesis of Pd/Pt core shell nanoparticles using area-selective atomic layer deposition. *Sci. Rep.* 5, 8470 (2015). (25) Feng, H., Elam, J. W., Libera, J. A., Stair, P. C. & Miller, J. T. Subnanometer palladium particles synthesized by atomic layer deposition. *ACS Catal.* 1, 665-673 (2011). (26) Lu, J. L., Fu, B. S., Kung, M. C., Xiao, G. M., Elam, J. W., Kung, H. H. & Stair, P. C. Coking and sintering-resistant palladium catalysts achieved through atomic layer deposition. *Science* 335, 1205-1208 (2012). (27) Lu, J. L., Liu, B., Greeley, J. P., Feng, Z. X., Libera, J. A., Lei, Y., Bedzyk, M. J., Stair, P. C. & Elam, J. W. Porous alumina protective coatings on palladium nanoparticles by self-poisoned atomic layer deposition. *Chem. Mater.* 24, 2047-2055 (2012)."

Reviewer #3 (Remarks to the Author):

The manuscript is written well. It proposes a strategy to allow independent control of Pt particle size and coverage for nano-sized supported particles using a combination of O₂-based and N₂ plasma-based atomic layer deposition (ALD). Using the ALD method to synthesize model Pt nanoparticles has a lot of advantages compared to the use of more classical methods. The ALD method is a precise deposition techniques with good control over e.g. conformality, thickness, and composition.

To the reviewers best knowledge the authors report independent tuning of size and coverage of Pt nanoparticles for the the first time. This is not only of great interest in the field of heterogeneous catalysis but also e.g. in surface science and micro-electronics. It could be speculated that a similar strategy can be used to deposit other (combinations of) metals.

The manuscript shows the independent control of size and coverage using a variety of techniques (GISAXS, XRD, SEM, HAADF-STEM) appropriate to characterize Pt particles.

The data presented in the manuscript supports the conclusions well.

Author reply:

We thank the reviewer for the careful evaluation of our work and for the useful suggestions to further improve the manuscript. The manuscript has been modified according to the specific comments of the reviewer, as explained below.

Putting the details of the GISAXS simulations in the supplementary information is a good decision. It would be beneficial to corroborate some of the assumptions in the GISAXS analysis. E.g. the validity of the assumption of a log-normal distribution function (line 46 p.3, Suppl. Inf.) and a relative width of $\sigma = 1.1$. The GISAXS simulations show good qualitative agreement with the experimental data; however a few cross-sections through the 2-dimensional data for e.g. constant q_y or q_z could also show a quantitative comparison (for figures S2, S3, S4, S5, S6). In this way the validity of the implicit choice of fixing the size distribution and fitting the shape (instead of the other way around) could be shown. As GISAXS is the main technique used to extract relevant parameters from the experiments, a quantitative comparison between simulations and experiments will strengthen the authors' conclusions. As all 2 dimensional data is available it will not generate much extra work for the authors to show a quantitative comparison.

Author reply:

As suggested by the reviewer, we have added 1D horizontal and vertical line profiles to the Supplementary Figures 7 to 11 (before S2 to S6). These graphs show that the q_y and q_z -positions of the main scattering peak are well reproduced and also the positions of the side minima and maxima in the experimental data and in the simulations are in good agreement, indicating that the average morphological parameters used for the calculations are correct.

The assumption of a lognormal distribution function for the particle size distribution is based on precedence in the literature, as is now explicitly mentioned in the Supplementary Information. Note that the average morphological parameters that are mentioned in the main text are determined without assuming a particle size distribution, as is now better explained in the Supplementary Information (e.g. Supplementary Figure 2). The distribution in particle sizes is only introduced to generate the simulated 2D patterns, which are compared to the experimental data in order to validate our GISAXS analysis strategy.

Changes to the manuscript:

Supplementary Figures 7 to 11 have been updated with additional graphs showing 1D line profiles through the GISAXS experimental data and simulations.

The following sentences have been added to the Supplementary Information: “To improve the agreement between simulation and experiment, the model that was used in step 2 to calculate the 1D line profiles is extended to account for the distribution in particle sizes. For the sake of simplicity, the particle height and width distributions are chosen to be coupled, in the sense that a distribution of particle radii at constant height/radius ratio implies also a distribution of particle heights. A lognormal distribution is assumed for the particle radius R, based on precedence in the literature⁷⁻¹⁰:

$$p(R) = \frac{1}{\sqrt{2\pi} R \ln(\sigma_R)} \exp\left(-\frac{1}{2} \left(\frac{\ln(R/\langle R \rangle)}{\ln(\sigma_R)}\right)^2\right) \quad (2)$$

with σ_R the dimensionless geometric standard deviation. The size distribution is kept equal for both types of particles in the model (full spheroids and hemispheroids).”

The following references have been added to the Supplementary Information: “(7) Blackman, J. A., Evans, B. L. & Maarouf, A. I. Phys. Rev. B 49, 13863-13871 (1994). (8) Söderlund, J., Kiss, L. B., Niklasson, G. A., Granqvist, C. G. Phys. Rev. Lett. 80, 2386-2388 (1998). (9) Kiss, L. B., Söderlund, J., Niklasson, G. A., Granqvist, C. G. Nanotechnology 10, 25-28 (1999). (10) Meshot, E. R., Verploegen, E., Bedewy, M., Tawfick, S., Woll, A. R., Green, K. S., Hromalik, M., Koerner, L. J., Philipp, H. T., Tate, M. W., Gruner, S. M. & Hart, A. J. ACS Nano 6, 5091-5101 (2012). (11) Qadri, M. U., Diaz Diaz, A. F., Cittadini, M., Martucci, A., Pujol, M. C., Ferré-Borrull, J., Llobet, E., Aguiló, M. & Díaz, F. Sensors 14, 11427-11443 (2014).”

A similar quantitative analysis of the SEM results (instead of the more limited qualitative analysis shown on page 15, line 256 and in figures 5b and 6b, will make the claims even more convincing.

Author reply:

As suggested by the reviewer, the SEM images in Figures 6b and 7b (before 5b and 6b) have been analyzed in a more quantitative way with respect to the particle size distribution and particle coverage.

Changes to the manuscript:

The following sentences have been added to the main text: “Indeed, **quantitative analysis of the SEM images** for samples *b*, *c* and *d* by manual counting the Pt nanoparticles in a 150 by 150 nm² area yields particle coverages of 1.58·10¹² cm⁻², 1.60·10¹² cm⁻² and 1.53·10¹² cm⁻², respectively. These values correspond to an average center-to-center distance $\langle D \rangle$ of 8.0 ± 0.1 nm. This value is higher than the one obtained from GISAXS analysis, 7.1 nm, which is likely due to the fact that agglomerated particles and particles at the edges of the SEM images are excluded from the particle count. For sample *a*, the contrast between the background and the small nanoparticles in SEM is insufficient to allow for a reliable particle count.”

The following paragraph has been added to the Supplementary Information: “As shown in Figure 8 in the main text, a good agreement is found between the average particle radius obtained from TEM analysis and the one derived from the GISAXS analysis. Supplementary Figure 12 below presents **additional analysis results for the SEM images included in Figure 6** of the main text, confirming again the agreement in average particle radius obtained from real-space electron microscopy measurements and reciprocal space GISAXS data. **The black lines for samples A, B and C are fitted lognormal functions to the particle size distributions.** The wide distribution observed for sample D is a consequence of the formation of wormlike structures when a large number of O₂-based ALD cycles is applied. For all lognormal fits, the value for the dimensionless geometric standard deviation σ_R is ~1.30. Similar fits to the size distributions obtained from TEM (Figure 8) yield a σ_R -value of ~1.25. Both of these values are larger than the value of 1.1 evaluated from GISAXS. However, for GISAXS simulations with a σ_R -value of 1.25 or 1.30, the scattering features are highly smoothed or damped, in disagreement with the experimental patterns. Similar differences in particle radius distribution obtained from TEM and GISAXS have been observed before for 1-10 nm Au nanoparticles embedded in a SiO₂ film and may be attributed to different sampling conditions.¹² For our SEM and TEM analyses, 300 to 1000 particles are measured from a small region of the sample (< 500 x 500 nm²) while GISAXS probes a sample area of ca. 300 nm x 2 cm, averaging over an estimated 10⁸ particles.”

Supplementary Figure 12 | SEM characterization of the Pt nanoparticle size distribution. Tuning the particle coverage by combining 0 (sample A), 20 (sample B), 30 (sample C) and 40 (sample D) O₂-based Pt ALD cycles with 60 (sample A), 40 (sample B), 30 (sample C) and 20 (sample D) N₂^{*}-based Pt ALD cycles. SEM images with 100 nm scale bars and derived particle size distributions. The black lines are fitted lognormal functions to the data.

The following reference has been added to the Supplementary Information: “(12) Sanchez, D. F., Marmitt, G., Marin, C., Baptista, D. L., Azevedo, G. M., Grande, P. L. & Fichtner, P. F. P. *Sci. Rep.* **3**, 3414 (2013).”

Therefore, I recommend publication with minor additions to the data analysis of GISAXS and SEM.

Reviewers' comments:

Reviewer #1 (Remarks to the Author):

The authors have addressed most of my concerns in a convincing way. However, not all points are fully finished. The still open questions or concerns are numbered based on the numbering from the first report:

5) The authors write that the idea to combine two different particle shapes in the simulations was based on a previous publication by Kaune et al. who used a model consisting of parallelepiped and spheroid particle geometries to describe the cluster shape of gold nanoparticles [ACS Appl. Mater. Interf., 1, 353, 2009]. As a consequence, I think that is necessary to mention such existing approach from literature in the main manuscript and to explain the ideas behind. Other readers will have the same problem in understanding the meaning of such approach with mixed wetting conditions. It is good scientific practice to cite all used references.

6) The reviewer agrees that calculation of error bars is a difficult task. However, error bars are very important in judging the quality of extracted parameters. I cannot accept the absence of error bars and the added simulations showing sensitivity give a good first hint but cannot replace error bars.

Reviewer #3 (Remarks to the Author):

The authors have given appropriate answers to the questions and remarks in my review and they have updated both the manuscript and the supplementary material, including figures. Also replies and changes to the manuscript based on input from the other referees has improved the manuscript.

To my opinion the manuscript can be published.

Reviewer #4 (Remarks to the Author):

This is an excellent study describing a method to control the size and spacing of Pt nanoparticles on a SiO₂ surface by ALD. The method combines two processes for ALD Pt: the first uses O₂ as the co-reactant, and produces a nanoparticle spacing that increases with increasing ALD Pt cycles, and the second uses N₂* as the co-reactant, and produces a constant

nanoparticle spacing. The O₂ process is used first to adjust the particle spacing and the N₂* process is used next to tune the Pt nanoparticle size. The authors primarily use in situ GISAXS to demonstrate this control, but they further support their findings using SEM and STEM.

I have no arguments regarding the science in the paper, and I find that the authors have done a thorough job of replying to the technical comments made by the three reviewers. My only criticism of this paper relates to novelty. In the initial review, Reviewer 2 expressed a similar concern as follows: "...the (only) novelty of the work is that the authors show that the use of a N₂* plasma as reactant allows for increasing the size of the Pt nanoparticles without changing the coverage of the Pt nanoparticles... this is a nice feature but by itself it does not warrant the publication of the work in Nature Communications...". For the most part, I agree with this statement. The effect of the N₂* plasma is the most significant finding, but there are some additional "firsts" in this paper that the authors point out below in their rebuttal to this comment. However, as I describe below, I do not feel that the N₂* process and the other accomplishments listed by the author meet the novelty criterion for publication in Nature Communications given that there are many previous publications that describe very similar work.

In response to this comment by Reviewer 2, and in defense of the novelty of their work, the authors present the following rebuttal (I have numbered the points):

"To stress the novelty of our work, we would like to respectfully remark that this work presents

- (1) an accurate tuning strategy to independently control the Pt nanoparticle size and coverage, even at high surface densities of nanoparticles for which precise control is often difficult to achieve due to easy merging and sintering of the Pt nanoparticles;
- (2) the first application of the N₂*-based Pt ALD process for the growth of Pt nanoparticles;
- (3) the first in situ characterization of the evolution in morphology during ALD of Pt and of noble metals in general, yielding insights in Pt particle ALD growth with a level of detail missing so far;
- (4) the first convincing experimental proof to date of the important role of atom and cluster surface diffusion during the commonly applied O₂-based Pt ALD process;
- (5) clear experimental evidence of the important role of the choice of reactant used in noble metal ALD."

There are a number of previous works, some of which were not referenced in the manuscript, that demonstrate to some degree each of these points as follows:

(1) Independent control over particle size and coverage has been demonstrated previously (refs. 21, 23). In these studies, control was achieved by decoupling ALD on the substrate from ALD on existing metal particles.

(2) The authors have used N₂* previously for Pt ALD films (ref. 37), and the extension to Pt nanoparticles is not surprising since Pt ALD nearly always produces nanoparticles during the nucleation stage.

(3) There have been numerous ex situ studies characterizing the evolution in morphology during ALD of Pt using GISAXS, XRF, and other methods (Christensen et al, Chem Mater, 21 516, 2009; Christensen et al, Small 5 (6) 750 2009,); Geyer et al, J Appl Phys 116 064905, 2014), and in situ studies of non-metal ALD processes (Klug et al, Rev Sci Instrum. 86, 113901, 2015).

(4) Diffusion of Pt species during O₂-based Pt ALD has been observed and discussed previously (Christensen et al, Chem Mater, 21 516, 2009)

(5) The important role of the choice of reactant used in noble metal ALD has been demonstrated previously for Pt and other noble metals (ref. 21).

Given these previous papers, I do not feel that the manuscript is sufficiently novel for publication in Nature Communications.

Responses to the reviewers' comments about the manuscript.

Reviewers' comments:

Reviewer #1 (Remarks to the Author):

The authors have addressed most of my concerns in a convincing way. However, not all points are fully finished. The still open questions or concerns are numbered based on the numbering from the first report:

Author reply:

We thank the reviewer for the careful evaluation of our work and his/her appreciation regarding most of our previous revisions and responses. We have addressed the two remaining comments of the reviewer in the second revised version of the manuscript as explained below.

5) The authors write that the idea to combine two different particle shapes in the simulations was based on a previous publication by Kaune et al. who used a model consisting of parallelepiped and spheroid particle geometries to describe the cluster shape of gold nanoparticles [ACS Appl. Mater. Interf., 1, 353, 2009]. As a consequence, I think that is necessary to mention such existing approach from literature in the main manuscript and to explain the ideas behind. Other readers will have the same problem in understanding the meaning of such approach with mixed wetting conditions. It is good scientific practice to cite all used references.

Author reply:

We agree with the reviewer that we should have added a reference to the work of Kaune et al.

Changes to the manuscript:

A reference to Kaune et al. has been added to the manuscript: "As exemplified in the Supplementary Information, best agreement with the experimental GISAXS patterns is obtained when a two particle model is used to describe the spheroidal particles."⁵³

(53) Kaune, G., Ruderer, M. A., Metwalli, E., Wang, W., Couet, S., Schlage, K., Röhlberger, R., Roth, S. V. & Müller-Buschbaum, P. ACS Appl. Mater. Interfaces 1, 353-360 (2009)."

The following sentence and reference have been added to the Supplementary Information to explain the origin of the idea behind the two-particle model used for the GISAXS simulations: "It should be noted that a similar simulation approach was used before by Kaune et al. who reported a two-particle model consisting of parallelepipeds and spheroids to reproduce both the intensity distribution of the side peaks and the interconnecting streaks observed in experimental GISAXS patterns recorded for gold cluster growth on poly(N-vinylcarbazole)."¹²

(12) Kaune, G., Ruderer, M. A., Metwalli, E., Wang, W., Couet, S., Schlage, K., Röhlberger, R., Roth, S. V. & Müller-Buschbaum, P. ACS Appl. Mater. Interfaces 1, 353 (2009)."

6) The reviewer agrees that calculation of error bars is a difficult task. However, error bars are very important in judging the quality of extracted parameters. I cannot accept the absence of error bars and the added simulations showing sensitivity give a good first hint but cannot replace error bars.

Author reply:

As requested by the reviewer, substantial effort was made to estimate the uncertainties of the extracted parameters (mean particle height $\langle H \rangle$, particle width $\langle W \rangle$ and center-to-center distance $\langle D \rangle$). The uncertainties were estimated based on a quantitative comparison of experimental and calculated 1D vertical (for $\langle H \rangle$) and horizontal (for $\langle W \rangle$ and $\langle D \rangle$) line profiles. The sum of squared residuals (SSR) was calculated for varying input values for one of the parameters (while the other parameters were kept constant). An increase in the SSR of ca. 50% (with respect to the SSR obtained when using the values extracted from our GISAXS analysis as input) was judged to give a good measure for the accuracy of our analysis approach. As such, by analyzing the SSR against a range of input values, the uncertainties of the extracted parameters were determined and added as error bars in Figure 3 of the manuscript.

Changes to the manuscript:

Error bars have been added to Figure 3 in the main text:

Figure 3 | Morphological evolution of Pt nanoparticles during ALD. In situ data on (a) mean center-to-center distance (top), particle width (middle) and particle height (bottom) against Pt loading for O₂-based Pt ALD (blue squares) and N₂*-based Pt ALD (green circles). The error bars represent the estimated uncertainties of the obtained values (all details can be found in the Supplementary Information). (b,c) Schematic representation of the GISAXS results for Pt loadings of (1) ~60, (2) ~120, and (3) ~190 Pt atoms / nm² obtained using O₂-based Pt ALD (b) and N₂*-based Pt ALD (c). (d) Pt dispersion, i.e. fraction of accessible Pt atoms, calculated from the particle dimensions and shape, as obtained from GISAXS, against Pt loading for O₂-based Pt ALD (blue squares) and N₂*-based Pt ALD (green circles).

The following section and figure have been added to the Supplementary Information: “To estimate the uncertainties of the obtained values for $\langle H \rangle$, $\delta\langle H \rangle$, experimental and simulated 1D vertical line profiles were compared and the sum of squared residuals (SSR) was calculated for varying values of $\langle H \rangle$ and fixed values of $\langle W \rangle$, $\sigma_R=1.1$, $\langle D \rangle$ and $\omega=0.4\langle D \rangle$ (i.e. the values extracted from the GISAXS analysis approach discussed above). The relative SSR was defined as the ratio between the SSR obtained at a certain $\langle H \rangle$ value and the SSR obtained at the $\langle H \rangle$ value extracted from our GISAXS analysis. Supplementary Figure 12a, middle shows the relative SSR for a range of values of $\langle H \rangle$ for growth stages corresponding to a Pt loading of ~ 155 atoms / nm² with the O₂-based (squares) and N₂*-based (circles) Pt ALD process, respectively. In both cases, a parabolic-type trend is observed and the extracted $\langle H \rangle$ value (red data point) is found near its minimum. Similar results were obtained for all $\langle H \rangle$ values plotted in Figure 3a, bottom in the main text. This confirms again the validity of our analysis approach to derive the average particle height. The uncertainties of the extracted $\langle H \rangle$ values were then estimated by determining the values of $\langle H \rangle$ for which the relative SSR increased to ca. 1.5 (green and blue data points). This increase in SSR yielded 1D vertical line profiles which deviated from the optimized simulated line profile (resulting from the analysis strategy discussed in the Method section above) and the experimental profile, as illustrated in Supplementary Figure 12a, right (O₂-based Pt ALD) and Supplementary Figure 12a, left (N₂*-based Pt ALD). The positions of the minima and maxima in the green and blue line profiles are clearly shifted with respect to the dashed vertical lines which indicate the extrema observed in the experimental 1D vertical line profiles. The uncertainties of the extracted $\langle H \rangle$ values were derived from the SSR analysis as indicated in Supplementary Figure 12a, middle and were added as error bars ($\pm\delta\langle H \rangle$) in Figure 3 in the main text.

A similar strategy was used to estimate the uncertainties of the obtained values for $\langle W \rangle$ and $\langle D \rangle$ (Supplementary Figures 12b and 12c, respectively). In both cases, SSR values were calculated based on the comparison of experimental and simulated 1D horizontal line profiles. In these profiles, the value of $\langle D \rangle$ mainly influences the q_y -position of the main scattering peak (via the interference function), while the q_y -positions of the minima and second maximum mainly originate from the form factor and thus the value of $\langle W \rangle$. Therefore, calculation of the SSR values for varying values of $\langle D \rangle$ was done by limiting the q_y -range of the 1D horizontal line profiles to the main scattering peak (Supplementary Figure 12c). In contrast, calculation of the SSR values for varying values of $\langle W \rangle$ was done by excluding the q_y -range of the main scattering peak from the 1D horizontal line profiles (Supplementary Figure 12b). For most of the $\langle W \rangle$ and $\langle D \rangle$ data points in Figure 3a, again a parabolic-type relation with a minimum near the extracted $\langle W \rangle$ and $\langle D \rangle$ values was found, allowing to estimate the uncertainty of these values by evaluating the simulations with a relative SSR of ca. 1.5, as explained above for the uncertainty of $\langle H \rangle$ and as illustrated in Supplementary Figures 12b and 12c, respectively. For the N₂*-based Pt ALD process and Pt loadings above ~ 160 atoms / nm², it was not possible to obtain a full parabola-type curve when varying $\langle D \rangle$ because of the constraint that $\langle D \rangle > \langle W \rangle$. In these cases, the uncertainty was estimated by evaluating only one simulation with a relative SSR of ca. 1.5 and doubling the obtained offset in $\langle D \rangle$.

Supplementary Figure 12 | Uncertainty estimation of extracted morphological parameters. The graphs in the middle present the relative sum of squared residuals (SSR) against the particle height (a), particle width (b) and center-to-center distance (c) obtained for selected growth stages of the O₂-based Pt ALD process (squares) and N₂*-based Pt ALD process (circles) corresponding to a Pt loading of ~155 atoms / nm². The graphs on the right and left display corresponding experimental (black data points) and calculated (red/green/blue curves) 1D vertical (a) and horizontal (b,c) line profiles for the O₂-based Pt ALD process and N₂*-based Pt ALD process, respectively. The horizontal line profiles are taken at the Si Yoneda position, i.e. $q_z = 0.722 \text{ nm}^{-1}$. The vertical line profiles are taken at the q_y -

position of maximum intensity. The dashed vertical lines indicate positions of minima and maxima in the experimental line profiles. The red curves are calculated using the values for $\langle D \rangle$, $\langle H \rangle$ and $\langle W \rangle$ that were obtained via the analysis procedure described in the Method section above (see red data points in the middle graphs). The green/blue curves correspond to an increase of the relative SSR to ca. 1.5 (see green/blue data points in the middle graphs). The estimated uncertainties $\delta\langle D \rangle$, $\delta\langle H \rangle$ and $\delta\langle W \rangle$ are indicated in the middle graphs and are added as error bars ($\pm\delta\langle D \rangle$, $\delta\langle H \rangle$ and $\delta\langle W \rangle$) in Figure 3 in the main text."

Reviewer #3 (Remarks to the Author):

The authors have given appropriate answers to the questions and remarks in my review and they have updated both the manuscript and the supplementary material, including figures.

Also replies and changes to the manuscript based on input from the other referees has improved the manuscript.

To my opinion the manuscript can be published.

Author reply:

We thank the reviewer for the careful evaluation of our work and recommendation for publication in *Nature Communications*.

Reviewer #4 (Remarks to the Author):

This is an excellent study describing a method to control the size and spacing of Pt nanoparticles on a SiO₂ surface by ALD. The method combines two processes for ALD Pt: the first uses O₂ as the co-reactant, and produces a nanoparticle spacing that increases with increasing ALD Pt cycles, and the second uses N₂^{*} as the co-reactant, and produces a constant nanoparticle spacing. The O₂ process is used first to adjust the particle spacing and the N₂^{*} process is used next to tune the Pt nanoparticle size. The authors primarily use in situ GISAXS to demonstrate this control, but they further support their findings using SEM and STEM.

I have no arguments regarding the science in the paper, and I find that the authors have done a thorough job of replying to the technical comments made by the three reviewers. My only criticism of this paper relates to novelty. In the initial review, Reviewer 2 expressed a similar concern as follows: "...the (only) novelty of the work is that the authors show that the use of a N₂^{*} plasma as reactant allows for increasing the size of the Pt nanoparticles without changing the coverage of the Pt nanoparticles... this is a nice feature but by itself it does not warrant the publication of the work in Nature Communications...". For the most part, I agree with this statement. The effect of the N₂^{*} plasma is the most significant finding, but there are some additional "firsts" in this paper that the authors point out below in their rebuttal to this comment. However, as I describe below, I do not feel that the N₂^{*} process and the other accomplishments listed by the authors meet the novelty criterion for publication in Nature Communications given that there are many previous publications that describe very similar work.

In response to this comment by Reviewer 2, and in defense of the novelty of their work, the authors present the following rebuttal (I have numbered the points):

"To stress the novelty of our work, we would like to respectfully remark that this work presents

- (1) an accurate tuning strategy to independently control the Pt nanoparticle size and coverage, even at high surface densities of nanoparticles for which precise control is often difficult to achieve due to easy merging and sintering of the Pt nanoparticles;
- (2) the first application of the N₂^{*}-based Pt ALD process for the growth of Pt nanoparticles;
- (3) the first in situ characterization of the evolution in morphology during ALD of Pt and of noble metals in general, yielding insights in Pt particle ALD growth with a level of detail missing so far;
- (4) the first convincing experimental proof to date of the important role of atom and cluster surface diffusion during the commonly applied O₂-based Pt ALD process;
- (5) clear experimental evidence of the important role of the choice of reactant used in noble metal ALD."

There are a number of previous works, some of which were not referenced in the manuscript, that demonstrate to some degree each of these points as follows:

- (1) Independent control over particle size and coverage has been demonstrated previously (refs. 21, 23). In these studies, control was achieved by decoupling ALD on the substrate from ALD on existing metal particles.

(2) The authors have used N_2^* previously for Pt ALD films (ref. 37), and the extension to Pt nanoparticles is not surprising since Pt ALD nearly always produces nanoparticles during the nucleation stage.

(3) There have been numerous ex situ studies characterizing the evolution in morphology during ALD of Pt using GISAXS, XRF, and other methods (Christensen et al, Chem Mater, 21, 516, 2009; Christensen et al, Small 5 (6), 750, 2008; Geyer et al, J Appl Phys 116, 064905, 2014), and in situ studies of non-metal ALD processes (Klug et al, Rev Sci Instrum. 86, 113901, 2015).

(4) Diffusion of Pt species during O_2 -based Pt ALD has been observed and discussed previously (Christensen et al, Chem Mater, 21 516, 2009).

(5) The important role of the choice of reactant used in noble metal ALD has been demonstrated previously for Pt and other noble metals (ref. 21).

Given these previous papers, I do not feel that the manuscript is sufficiently novel for publication in Nature Communications.

Author reply:

We thank the reviewer for the careful evaluation of our work and his/her appreciation regarding the science in the paper. However, the reviewer was not convinced by our previous response to Reviewer #2 concerning the novelty of our work. The reviewer mentioned that the novelty criterion is not met because there are previous works which are to some degree related to the novelty claims that we listed in our rebuttal. While we definitely recognize the scientific value of the previous works cited by the reviewer, we still feel that our work presents substantial novel contributions concerning scientific results as well as methodology. Please find below a point-by-point response. For each point, I present the novelty claimed in our previous rebuttal, the reply of the reviewer and our new response:

(1) This work presents an accurate tuning strategy to independently control the Pt nanoparticle size and coverage, even at high surface densities of nanoparticles for which precise control is often difficult to achieve due to easy merging and sintering of the Pt nanoparticles.

Reviewer reply: Independent control over particle size and coverage has been demonstrated previously (refs. 21, 23). In these studies, control was achieved by decoupling ALD on the substrate from ALD on existing metal particles.

Author reply: References 21 and 23 report the growth of bimetallic nanoparticles using the ALD method. Weber et al. (ref. 23) demonstrated the synthesis of Pt/Pd and Pd/Pt core/shell nanoparticles, while Lu et al. (ref. 21) reported the fabrication of Pt/Pd alloys as well as of Pt/Pd, Pt/Ru and Pd/Pt core/shell nanoparticles. In both works, the core/shell nanoparticles are obtained by optimization of ALD process conditions to obtain (1) growth of core nanoparticles on an Al_2O_3 support, followed by (2) selective growth of the shell material on the deposited metal cores. While the presented strategies provide control over the size of the metal cores and shell thicknesses, the authors did not show how the particle coverage can be tuned. In fact, the particle coverage was only considered in ref. 23, where Weber et al. reported values obtained from HAADF-STEM images to prove that adding a Pt shell onto Pd cores does not affect the NP coverage considerably. Therefore, we cannot agree with the statement of the reviewer that independent control over particle size AND

coverage has been demonstrated previously. We are confident that the submitted manuscript presents a novel contribution concerning the tuning of both particles sizes AND particle coverages via ALD.

(2) This work presents the first application of the N_2^* -based Pt ALD process for the growth of Pt nanoparticles.

Reviewer reply: The authors have used N_2^ previously for Pt ALD films (ref. 37), and the extension to Pt nanoparticles is not surprising since Pt ALD nearly always produces nanoparticles during the nucleation stage.*

Author reply: Although the N_2^* -based Pt ALD process was already reported in 2012 for the growth of Pt films (ref. 37), it has so far not been considered by the ALD community for its use in nanoparticle deposition. In this work, we demonstrate that the nanoparticle growth mode is considerably different for the N_2^* -based Pt ALD process compared to the standard O_2 -based process: higher nanoparticle densities can be obtained for higher Pt loadings. Although the growth of nanoparticles with the N_2^* -based Pt ALD process in itself is not too surprising, as mentioned by the reviewer, we believe that the very different nanoparticle growth mode revealed in the submitted manuscript is a very surprising result that will trigger other researchers to further investigate the N_2^* -based Pt ALD process for Pt nanoparticle ALD growth.

(3) This work presents the first *in situ* characterization of the evolution in morphology during ALD of Pt and of noble metals in general, yielding insights in Pt particle ALD growth with a level of detail missing so far.

Reviewer reply: There have been numerous ex situ studies characterizing the evolution in morphology during ALD of Pt using GISAXS, XRF, and other methods (Christensen et al, Chem Mater, 21, 516, 2009; Christensen et al, Small 5 (6), 750, 2008; Geyer et al, J Appl Phys 116, 064905, 2014), and in situ studies of non-metal ALD processes (Klug et al, Rev Sci Instrum. 86, 113901, 2015).

Author reply: Although the interest in synchrotron-based characterization of ALD-grown materials has increased in recent years, the number of studies exploiting these methods is still rather limited. This is certainly the case for *in situ* investigations of ALD processes at synchrotrons as these require dedicated setups and beamlines (e.g. Devloo-Casier et al, J Vac Sci Technol A 32, 010801, 2014; Geyer et al, Rev Sci Instrum 85, 055116, 2014; Klug et al, Rev Sci Instrum 86, 113901, 2015; ref. 48). To the best of our knowledge, we are the only group so far who designed a dedicated setup that enables *in situ* synchrotron-based characterizations during plasma-enhanced ALD processes (ref 48), which was an absolute requirement for the present study.

As listed by the reviewer, there are indeed a number of *ex situ* studies that used synchrotron-based X-ray scattering methods to study the morphology of Pt nanoparticles deposited by the standard Pt ALD process using O_2 gas:

- *Christensen et al, Chem Mater 21, 516, 2009* describes a study of the initial nucleation of Pt ALD on planar $SrTiO_3(001)$ substrates using *ex situ* XRF and GISAXS measurements. While Christensen

et al. based their analysis on a couple of GISAXS patterns and XRF data points (for Pt loadings up to ~ 250 atoms/nm²), the results presented in the submitted work are based on the analysis of a continuous set of *in situ* GISAXS patterns and XRF spectra, offering a greater level of detail on the morphological evolution. Christensen et al extracted data from the GISAXS patterns by model fitting to the one-dimensional horizontal and vertical line profiles. In the submitted manuscript, the full 2D patterns are simulated, yielding more insights in the particle shape, revealing in turn a difference in particle wetting conditions for the O₂-based and N₂^{*}-based Pt ALD processes. Finally, the capability of performing *in situ* measurements allowed us to provide convincing evidence of the power of the proposed tuning strategy.

- *Christensen et al, Small 5, 750, 2009* presents an *ex situ* study of Pt ALD on SrTiO₃ powder particles (nanocubes) using SAXS, WAXS and XAS measurements. Samples modified with 1, 2, 3, 4 and 5 ALD cycles were investigated. The interparticle distance and particle size were shown to increase with increasing number of ALD cycles, in line with our results. A direct comparison with our work is, however, not straightforward because the *Small* paper concerns depositions on a high surface area support using much higher precursor doses than applied in our work. Therefore, larger particle sizes are obtained for much lower number of ALD cycles.
- *Geyer et al, J Appl Phys 116, 064905, 2014* used *ex situ* GISAXS measurements to study the lateral growth of Pt islands grown on Si wafers with 2nm native SiO₂. The horizontal line profile was analyzed for 4 measurements corresponding to 30, 40, 50 and 60 ALD cycles, respectively. To interpret the GISAXS data, a continuous nucleation model was assumed in which new nuclei are formed during each ALD cycle on the available surface area of the substrate. This model resulted in a distribution of island sizes marked by a maximum island size. Plotting this maximum island size against the number of ALD cycles yielded a lateral growth rate of 0.51 Å/cycle. The GISAXS analysis strategy used by Geyer et al. differs from the approach used in the submitted manuscript. Based on our GISAXS, TEM and SEM results, the particle growth seems not to be dominated by continuous nucleation but by diffusion mediated particle coalescence, yielding a particle size distribution concentrated around a characteristic average particle size. This difference could be related to a different O₂ exposure; however, the O₂ pressure and pulse time were not specified in the work by Geyer et al. It should also be noted that a much more profound GISAXS analysis, including full simulations of 2D GISAXS patterns, was performed in the submitted manuscript than in the work of Geyer et al.

In addition to the above discussed publications that were also mentioned by the reviewer, there are also few *in situ* XAS studies of Pt ALD processes [ref 39; Filez et al, Catal Today 229, 2, 2014]. These works provided interesting insights in the evolution in the Pt oxidation state during the initial island growth.

Based on the overview presented above, it is clear that the *in situ* nature of the XRF and GISAXS data and the detailed analysis presented in the submitted manuscript definitely provide novel insights in the nucleation of Pt particles during ALD, over previous works where the analysis was based on *ex situ* GISAXS or SAXS data.

(4) This work presents the first convincing experimental proof to date of the important role of atom and cluster surface diffusion during the commonly applied O₂-based Pt ALD process.

Reviewer reply: Diffusion of Pt species during O₂-based Pt ALD has been observed and discussed previously (Christensen et al, Chem Mater, 21, 516, 2009).

Author reply: We agree with the reviewer that diffusion of Pt species during O₂-based Pt ALD was discussed in the work by Christensen et al, Chem Mater, 21, 516, 2009. However, there are important differences to note with respect to the results presented in the submitted manuscript:

- The first difference concerns the substrate onto which the Pt was deposited. Christensen et al. studied the growth on SrTiO₃, while the submitted work concerns the growth on a SiO₂ surface. The XRF data in both works reveal a clear difference in substrate-related growth behavior: while a substrate-enhanced growth was observed on SrTiO₃, a delayed growth was observed on SiO₂, marked by a very slow initial growth rate (Supplementary Figure 1). The observation of substrate-enhanced growth in the paper of Christensen et al. lead to a proposed growth model that includes preferred deposition of Pt onto the SrTiO₃ followed by diffusion of these Pt species to the growing nanoparticles. In addition to the growth model proposed by Christensen et al., the submitted manuscript shows that diffusion of Pt species also plays an important role in systems where growth on the substrate is not accelerated.
- A second difference concerns the experimental data presented and analyzed, i.e. the experimental proof. As mentioned above, the conclusions from GISAXS in the paper of Christensen et al. were based on the analysis of 4 patterns, i.e. for Pt nanoparticles grown using 10, 20, 30 and 40 ALD cycles on planar SrTiO₃ (001) substrates. The authors reported a decrease in center-to-center distance, which was explained by particle coalescence. Christensen et al. also showed corresponding SEM images. From these, it is clear that for 30 and 40 ALD cycles worm-like Pt structures are formed instead of isolated nanoparticles. These worm-like features may arise due to overlap of two adjacent Pt nanoparticles, i.e. static coalescence of nanoparticles. Therefore, the experimental data presented by Christensen et al. do not necessarily provide a solid proof for the occurrence of dynamic (diffusion-mediated) coalescence. In contrast, the submitted work presents the analysis results of an *in situ* recorded data set of GISAXS patterns in the regime where isolated Pt nanoparticles are formed on the SiO₂ surface. The fact that the center-to-center distance gradually decreases with increasing Pt loading, even in the regime of isolated Pt nanoparticles, can only be explained by dynamic (diffusion-mediated) coalescence. As such, the data presented in the submitted manuscript provide the first convincing evidence for the important role of Pt atom and cluster diffusion during the very early growth stages of Pt ALD.

Changes to the manuscript: A reference to the paper by Christensen et al. has been added to the manuscript.

(5) This work presents clear experimental evidence of the important role of the choice of reactant used in noble metal ALD.

Reviewer reply: The important role of the choice of reactant used in noble metal ALD has been demonstrated previously for Pt and other noble metals (ref. 21).

Author reply: As mentioned above, reference 21 reports the synthesis of bimetallic core/shell nanoparticles. To achieve deposition of the noble metal cores, ALD process conditions need to be selected for which nucleation occurs on the metal oxide support. Subsequently, the ALD process conditions for depositing the shell material need to be optimized such that selective growth on the noble metal cores is achieved. In the work by Lu et al. this selectivity is tuned by changing the type of reactant. For the case of Pt ALD, for example, nucleation on oxide surfaces can be achieved at 150°C by using O₃ as reactant, while selective growth on noble metals can be obtained by using O₂ gas at 150°C. Reference 21 therefore demonstrates the importance of the choice of reactant for (non-)selective noble metal ALD. On the other hand, in the submitted manuscript, the choice of reactant is shown to have a considerable influence on the initial island growth mode and the resulting nanoparticle coverage, size and shape.

REVIEWERS' COMMENTS:

Reviewer #1 (Remarks to the Author):

The authors have addressed all my concerns in a convincing way and I can recommend publication in the present format. The manuscript clearly has sufficient novelty to justify publication.